# Host population dynamics influence *Leptospira* spp. transmission patterns among *Rattus norvegicus* in Boston, Massachusetts, US

Nathan E. Stone[1], Camila Hamond[2], Joel R. Clegg[1], Ryelan F. McDonough[1], Reanna M. Bourgeois[1], Rebecca Ballard[1], Natalie B. Thornton[1], Marianece Nuttall[1], Hannah Hertzel[3], Tammy Anderson[2], Ryann N. Whealy[1,4], Skylar Timm[1,4], Alexander K. Roberts[1,4], Verónica Barragán[5], Wanda Phipatanakul[6], Jessica H. Leibler[7], Hayley Benson[3], Aubrey Specht[3], Ruairi White[3], Karen LeCount[2], Tara N. Furstenau[4], Renee L. Galloway[8], Nichola J. Hill[9], Joseph D. Madison[9,10], Viacheslav Y. Fofanov[1,4], Talima Pearson[1], Jason W. Sahl[1], Joseph D. Busch[1], Zachary Weiner[8], Jarlath E. Nally[11], David M. Wagner[1�]*, Marieke H. Rosenbaum [3�]*

1 The Pathogen and Microbiome Institute, Northern Arizona University, Flagstaff, Arizona, United States of America, 2 National Veterinary Services Laboratories, APHIS, United States Department of Agriculture, Ames, Iowa, United States of America, 3 Department of Infectious Disease and Global Health, Cummings School of Veterinary Medicine, Tufts University, North Grafton, Massachusetts, United States of America, 4 School of Informatics, Computing, and Cyber Systems, Northern Arizona University, Flagstaff, Arizona, United States of America, 5 Universidad San Francisco de Quito, Colegio de Ciencias Biologicas y Ambientales, Quito, Ecuador, 6 Division of Allergy and Immunology, Boston Children's Hospital, Boston, Massachusetts, United States of America, 7 Department of Environmental Health, Boston University School of Public Health, Boston, Massachusetts, United States of America, 8 Bacterial Special Pathogens Branch, Centers for Disease Control and Prevention, Atlanta, Georgia, United States of America, 9 Biology Department, University of Massachusetts Boston, Boston, Massachusetts, United States of America, 10 Center for Conservation Genomics, Smithsonian's National Zoo and Conservation Biology Institute, Washington, D.C., United States of America, 11 Infectious Bacterial Diseases Research Unit, ARS, United States Department of Agriculture, Ames, Iowa, United States of America

☉ These authors contributed equally to this work.
* marieke.rosenbaum@tufts.edu (MHR); dave.wagner@nau.edu (DMW)

## Abstract

Leptospirosis (caused by pathogenic bacteria in the genus *Leptospira*) is prevalent worldwide but more common in tropical and subtropical regions. Transmission can occur following direct exposure to infected urine from reservoir hosts, or a urine-contaminated environment, which then can serve as an infection source for additional rats and other mammals, including humans. The brown rat, *Rattus norvegicus*, is an important reservoir of *Leptospira* spp. in urban settings. We investigated the presence of *Leptospira* spp. among brown rats in Boston, Massachusetts and hypothesized that rat population dynamics in this urban setting influence the transportation, persistence, and diversity of *Leptospira* spp. We analyzed DNA from 328 rat kidney samples collected from 17 sites in Boston over a seven-year period (2016–2022); 59 rats representing 12 of 17 sites were positive for *Leptospira* spp. We used 21 neutral microsatellite loci to genotype 311 rats and utilized the resulting data to investigate genetic connectivity among sampling sites. We generated whole genome sequences

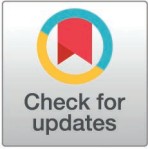 OPEN ACCESS

**Data availability statement:** MSAT multi-locus genotypes for 311 R. norvegicus are provided in S2 Table. Whole genome sequencing reads have been deposited in GenBank under BioProject PRJNA1080226 with SRA accession numbers SRR28374870-SRR28374875, SRR28374877-SRR28374878, SRR28374880, SRR28374882-SRR28374900 for isolate genomes and SRR28374876, SRR28374879, SRR28411353-SRR28411357, SRR28411359-SRR28411365, SRR28411370 for enriched genomes. Raw AmpSeq data for up to 42 loci are also provided in GenBank under the same BioProject with SRA accession numbers SRR28374881, SRR28411338-SRR28411352, SRR28411358, SRR28411366-SRR28411369 and SRR28765937- SRR28765945. Custom python script can be found at [https://gist.github.com/jasonsahl/2a232947a3578283f54c](https://gist.github.com/jasonsahl/2a232947a3578283f54c).

**Funding:** This research was supported in part by an appointment to the Animal and Plant Health Inspection Service (APHIS) Research Participation Program administered by the Oak Ridge Institute for Science and Education (ORISE) through an interagency agreement between the U.S. Department of Energy (DOE) and the U.S. Department of Agriculture (USDA). ORISE is managed by ORAU under DOE contract number DE-SC0014664 (to CH). This project was also funded in part by the National Institute of Allergy and Infectious Diseases in the National Institutes of Health (NIH) award number R01AI172924 (to TP), the National Institute of Environmental Health Sciences under award number K08ES035460 (to MHR), the National Center for Advancing Translational Sciences, NIH under award number KL2TR002545 (to MHR, to HH, to HB and to AS), and the National Institute of Allergy and Infectious Diseases, NIH, award number K-24 AI 106822 (to WP). Work conducted by JDM was completed while supported by the NSF Postdoctoral Research Fellowships in Biology Program, Award No. 1907311 (to JDM). The funders had no role in study design, data collection and analysis, decision to publish, or preparation of the manuscript.

**Competing interests:** The authors have declared that no competing interests exist.

for 28 *Leptospira* spp. isolates obtained from frozen and fresh tissue from some of the 59 positive rat kidneys. When isolates were not obtained, we attempted genomic DNA capture and enrichment, which yielded 14 additional *Leptospira* spp. genomes from rats. We also generated an enriched *Leptospira* spp. genome from a 2018 human case in Boston. We found evidence of high genetic structure among rat populations that is likely influenced by major roads and/or other dispersal barriers, resulting in distinct rat population groups within the city; at certain sites these groups persisted for multiple years. We identified multiple distinct phylogenetic clades of *L. interrogans* among rats that were tightly linked to distinct rat populations. This pattern suggests *L. interrogans* persists in local rat populations and its transportation is influenced by rat population dynamics. Finally, our genomic analyses of the *Leptospira* spp. detected in the 2018 human leptospirosis case in Boston suggests a link to rats as the source. These findings will be useful for guiding rat control and human leptospirosis mitigation efforts in this and other similar urban settings.

## Author summary

Leptospirosis is a common zoonotic disease caused by *Leptospira spp.* bacteria for which urban rats are a known reservoir. We tested 328 rats across Boston for *Leptospira* spp. and analyzed the results in relation to the genetic population structure of rats and found that rat population dynamics influence *Leptospira* spp. transmission in Boston, US. We also found evidence of a strain of *Leptospira* isolated from rats that closely matches a strain isolated from a human with leptospirosis in the area. Taken together, our results highlight the importance of understanding rat ecology, population structure and movement in reducing risk of *Leptospira* spp. transmission to novel rat populations and to humans.

## Introduction

Rats (*Rattus norvegicus* and *R. rattus*) are highly invasive species globally, persisting on all continents except Antarctica and in most habitats [1,2]. They cause devastating impacts to natural ecosystems [3] and are considered a serious public health threat [4,5]. *Rattus norvegicus* (the brown rat) is ubiquitous in urban environments [6,7] and has been documented to carry/transmit (either directly or indirectly) an abundance of bacterial and viral pathogens in urban ecosystems, including *Leptospira* spp., *Bartonella* spp., *Borrelia* spp., *Yersinia pestis*, *Streptobacillus moiliformis*, antibiotic-resistant *Staphylococcus* spp., *Shigella* spp., *Campylobacter* spp., *Escherichia coli*, *Rickettsia* spp., *Salmonella* spp., influenza A virus, and Seoul hantavirus [8–11].

Multiple genetic studies utilizing various methods have assessed *R. norvegicus* population structure, movements, and dispersal patterns in urban settings in both temperate and tropical climates [2,12–17], consolidated in a recent review [6].

Despite this somewhat limited body of research, findings are generally uniform across all cities assessed (Vancouver, Canada; New York City, US; New Orleans, US; Baltimore, US; Salvador, Brazil; and Hauts-de-Seine, France): 1) individual *R. norvegicus* utilize small home ranges, display site fidelity, and display philopatry to their birth site; and 2) dispersal events are more common in adult rats (male or female), but rare overall, and often inhibited by major roads and rivers, which act as dispersal barriers.

Leptospirosis is the most globally widespread bacterial zoonoses resulting in over 1 million human infections annually and nearly 60,000 deaths [18,19]. Brown rats are asymptomatic, chronic carriers of pathogenic *Leptospira* spp. [20–23] and the most important known source of human leptospirosis infections [24]. Leptospires colonize and replicate in the proximal renal tubules of rats [20–22], with transmission occurring after leptospires are shed through urine, either via direct contact with the urine or indirectly via environmental or fomite contamination [24]. Globally, > 30% of *R. norvegicus* are estimated to be infected with *Leptospira* spp. [25].

Understanding how rat movements relate to *Leptospira* spp. transmission in urban settings is of vital importance to guide the rational design of disease mitigation strategies. A recent study of rat population structure and pathogen density in an urban setting (Vancouver) suggested *L. interrogans* prevalence was associated with rat population connectivity [16], wherein related rats sampled from neighboring blocks shared the same pathogen status (*i.e.*, infected/not infected). Our study builds upon this finding by pairing fine-scale population genetic analyses of rats in another urban city (Boston, US) with fine-scale genomic analyses of the *Leptospira* spp. they carry and transmit.

Historically, the acquisition of *Leptospira* spp. genomes from reservoir (*e.g.*, rats) and incidental (*e.g.*, humans) hosts has relied on the ability to obtain cultured isolates, which is inherently challenging [26]. Obtaining isolates from human leptospirosis cases is further inhibited by the administration of antibiotics used to treat the disease [27], after which isolation of live leptospires is almost impossible. As such, genomic analyses of infecting strains are lacking, and these data are critical for improved understanding of *Leptospira* spp. ecology and epidemiology. We implemented two new culture-independent techniques for the genetic/genomic characterization of infecting *Leptospira* spp. in rats and humans in Boston: DNA capture and enrichment [28] and amplicon sequencing (AmpSeq; described herein). Our goal was to gain a deeper understanding of the nuanced *Leptospira* spp. transmission network among urban *R. norvegicus* and potential spillover into humans. Our findings are useful for guiding rat control and human leptospirosis mitigation efforts in urban settings.

## Results

### *lipL32* qPCR in rodent samples

In total, 59 *R. norvegicus* kidney samples and one fresh urine sample (derived from an animal that also had a positive kidney sample) were *lipL32* PCR positive, revealing an overall *Leptospira* spp. infection rate of 18.0% (59 of 328 animals) in this *R. norvegicus* dataset (S1 Table). *Leptospira*-positive kidneys included 37 frozen samples (17.6%; 37/210) yielding an average Ct value of 28.77 ± 5.58, and 23 fresh samples from 22 rats (18.6%; 22/118) yielding an average Ct value of 27.1 ± 6.06. Four bycatch mice and one squirrel were *lipL32* PCR negative (S2 Table). Fifty-six of the infected *R. norvegicus* were detected among 20/33 collections representing 12/17 sampling sites (Figs 1 and S1, and S1 Table) and positive rats were detected in all sampling years [2016 = 3/42 (7.1%), 2017 = 6/50 (12.0%), 2018 = 16/68 (23.5%), 2019 = 9/42 (21.4%), 2020 = 3/8 (37.5%), 2021 = 14/70 (20.0%), 2022 = 8/48 (16.7%)] (S1 Fig). Precise locations were unknown for infected *R. norvegicus* collected in 2020 (samples B1, B2, and B4; S1 and S2 Tables). Infected rats were more commonly adults (58/59) than juveniles (1/59) ($X^2 = 60.68$, $p < 0.001$), and a higher proportion of infected rats were collected in fall (0.277) and winter (0.283) compared to spring (0.196) and summer (0.123); a chi-square test indicated a non-random association between *Leptospira* spp. infection and season ($X^2 = 10.33$, $p < 0.05$). We also detected a significant association between *Leptospira* spp. infection and site ($X^2 = 26.58$, $p < 0.05$); however, this association is likely driven by capture of a single rat from site 4 that was infected, and 3/5 infected rats captured at site 15 (60%); all other sites ranged from 0-33.3%

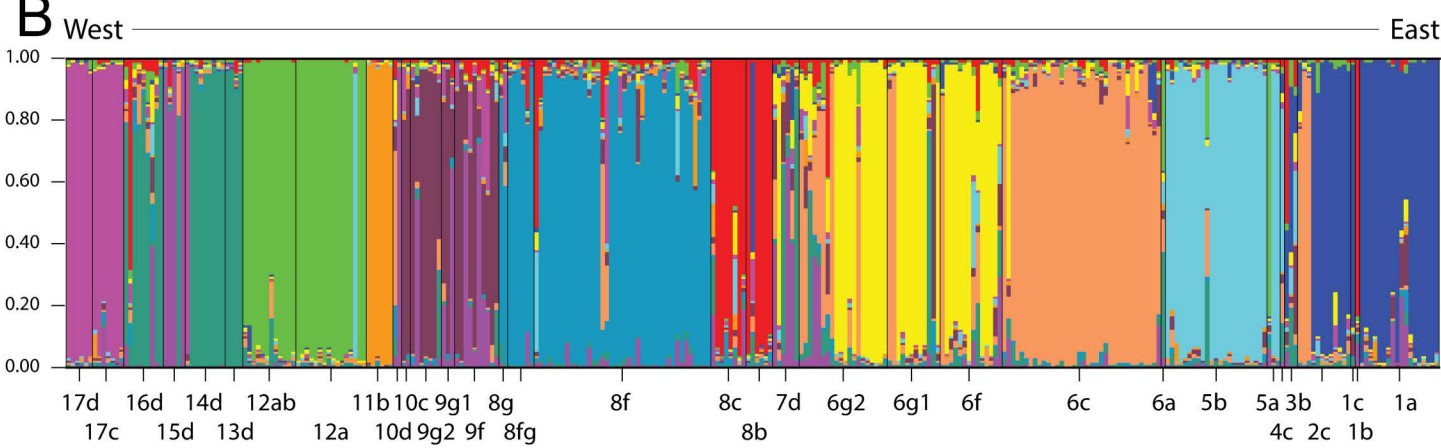

**Fig 1. *Rattus norvegicus* population assignment.** STRUCTURE population assignment of 311 *R. norvegicus* with collections color coded according to genetic group (S1 Table); likely dispersal barriers and major open spaces are indicated with black bars and green shading, respectively. A) Inferred migrants are represented by one-sided arrows; shared genetic groups among collections from sites 13, 14, and 16, and separately sites 9 and 10, are indicated by thin lines; and *lipL32* PCR positivity at 12 of 17 sites is indicated with asterisks. Two inferred migrants (sites 2 and 6) were both *lipL32* PCR positive and are denoted by asterisks on one-sided arrows. B) Each individual *R. norvegicus* (n = 311) is assigned a probability of membership ($Q ≥ 0.75$) to one of 12 genetic groups (S3 Table). Collection ID is indicated on the X-axis and probability ($Q$) on the Y-axis; each thin vertical line represents the $Q$ value for an individual rat. The map in panel A was created using ArcGIS software by Esri. ArcGIS and Arc-Map are the intellectual property of Esri and are used herein under license. Copyright Esri. All rights reserved. For more information about Esri software, please visit www.esri.com. Basemap: Light Gray Canvas Base https://www.arcgis.com/home/item.html?id=8b3d38c0819547faa83f7b7aca80bd76.

positivity. No other significant associations were observed between other sampling variables (sex, sampling year, genetic group) and the detection of *Leptospira* spp.

## Culture

Twenty-eight isolates were obtained from *lipL32* PCR positive kidneys, including 15 from 37 frozen samples and 13 from 22 fresh samples. Nearly half of the collections (15/33) are represented by these isolates as well as all sampling years (2016 = 1, 2017 = 2, 2018 = 6, 2019 = 3, 2020 = 3, 2021 = 7, 2022 = 6; S2 Table).

## Serotyping of *Leptospira* spp. isolates

All isolates had high agglutination titers with reference antiserum specific for serogroup Icterohaemorrhagiae (S2 Table). Isolates were completely seronegative (at 1:100 or above) [29] for any other serogroup tested, except for three samples: MAR_57 had a titer of 1:102,400 against Icterohaemorrhagiae and 1:100 against Pyrogenes; Cad103 had a titer of 1:51,200 against Icterohaemorrhagiae, 1:200 against Djasiman, and 1:100 against Hardjo; and R304 had a titer of 1:52,400 against Icterohaemorrhagiae and 1:100 against Pyrogenes. We consider all three of these isolates to be serogroup Icterohaemorrhagiae as it is not uncommon to see cross reactions with other serogroups [30].

## Evaluation of virulence

Intraperitoneal inoculation of all hamsters with $10^8$ leptospires derived from a frozen *R. norvegicus* kidney (strain R47) resulted in acute disease requiring all hamsters to be euthanized four days post-infection. Liver and kidney samples from each infected hamster were positive by both culture and *lipL32* qPCR; Ct values for positive liver and kidney samples were on average 19.04 ± 0.63 and 23.64 ± 0.75, respectively.

## Rat MSAT results

**MSAT marker validation.** Twenty-one microsatellite (MSAT) markers were suitable for use in this population genetic analyses of *R. norvegicus*. Only one marker (D12Rat76) deviated from Hardy-Weinberg equilibrium in one collection (8f) at the Bonferroni corrected $a = 0.000595$ ($a = 0.05/84$ tests) and no signatures of linkage disequilibrium were detected. Null alleles, large allelic dropout, or scoring errors due to stuttering also were not detected, although we did detect excessive homozygosity in two markers (D12Rat76 and D5Rat83) for collection 2c and one marker in four other collections (D5Rat33 in 5b, D10Mit5 in 6g1, D7Rat13 in 12a, and D1Cebr3 in 12ab). No single marker displayed excessive homozygosity in more than one collection. To check for allele scoring errors, we replicated PCRs for all 21 loci across 94 *R. norvegicus* DNAs (30.2% of the final dataset) and detected an allele scoring error rate of 0.23% (9 errors from 3,948 possible allele calls).

**Population genetic analyses: Determination of genetic groups and population assignment using STRUCTURE.** The delta-*K* method estimated 2 and 12 genetic groups as the most likely number of populations sampled in the 311 individual rats from 33 collections/17 sites (S2A Fig). The *K* of 12 is more consistent with what is known about the philopatric nature of *R. norvegicus* [6,12] and physical distances among our collection sites. Variance among the L(K) values is very low at *K* = 12 but rapidly increases beyond this number (S2B Fig); this phenomenon is discussed in the STRUCTURE manual and can be expected to occur after the real *K* is reached [31]. Furthermore, the higher delta-*K* value at *K* = 2 could simply be that *K* = 1 is consistently a very poor solution and the model improved dramatically between *K* = 1 and 2. The 12 genetic groups were each assigned an arbitrary color in STRUCTURE (Fig 1) and that color was retained across all analyses, figures, and tables. Thus, we refer to the genetic groups by the following color designations: Blue, Brown, Evergreen, Green, Light Blue, Orange, Pink, Purple, Red, Steel, Tan, Yellow, and admixed (indicated by the color White) (Fig 1 and S1 Table). To aid in visualizing patterns across space and time, collections were color coded according to the most dominant genetic group (≥50%) present in that collection (defined in S1 Table and displayed in Figs 1, 2 and

S1). In total, 247 rats were assigned to one of these genetic groups at $Q \geq 0.75$ (see Materials and Methods), whereas 64 rats were assigned to the admixed group (S3 Table). Due to repeated sampling over multiple years at sites 1, 5, 6, 8, 9, 10, 12, and 17, we detected stable rat populations at 7/8 sites for up to five years: site 1 (Blue group: 2016–2018), site 5 (Light Blue group: 2016–2017), site 6 (Tan group: 2018–2022 and Yellow group: 2021–2022), site 8 (Red group: 2017–2018 and Steel group: 2021–2022), site 9 (Brown group: 2021–2022), site 12 (Green group: 2016–2017), and site 17 (Pink group: 2018–2019); as well as population turnover/invasion followed by persistence at two sites: site 6 (Tan group in 2018 to Yellow and Tan groups from 2020-2021) and site 8 (Red group from 2017-2018 to Steel group from 2021-2022) (Figs 1 and S1).

Using $Q$ values ≥ 0.75, we inferred potential *R. norvegicus* migrations as indicated by the trapping of a rat assigning to a distinct genetic group among a collection of rats belonging to a different genetic group; any highly admixed individuals were not treated as migrants due to the potential for mis-assignment. In total, we inferred eleven such migration events, seven of which were adult rats (5 females, 2 males) and four were juveniles (1 female, 3 males). Rat movement distances ranged from ~0.8-5.3 km. Migrations were suspected from collection 17c or 17d to collection 14d (1 juvenile female, $Q=0.93$); collection 6c to collection 2c (2 juveniles males, $Q=0.91$ and 0.95, and 1 adult male, $Q=0.76$); collection 13d, 14d, or 16d (all belonging to the Evergreen genetic group) to collection 15d (1 adult female, $Q=0.81$); collection 5a or 5b to collection 12a (1 adult female, $Q=0.92$); collection 9g1, 9g2, or 10c (Brown genetic group) to collection 6g1 (1 adult male, $Q=0.76$); collection 15d to collection 10d (1 adult female, $Q=0.97$); collection 12a or 12b to collection 5a (1 juvenile male, $Q=0.82$); collection 8b or 8c to collection 1b (1 adult female, $Q=0.83$); and collection 13d, 14d, or 16d (Evergreen genetic group) to collection 7d (1 adult female, $Q=0.87$) (Figs 1 and S1, and S3 Table). Two inferred migrants (R335 from collection 2c and Cad89 from 6g1) were *lipL32* positive (Fig 1 and S3 Table). Some suspected migrations could also result from home range overlap among neighboring collections.

**Population genetic analyses: Genetic structure.** Principal Coordinates Analysis (PCoA) generally supported the 12 *R. norvegicus* genetic groups identified by STRUCTURE and indicated two genetic groups (Blue and Steel) may be more genetically isolated from the other ten groups (S3 Fig). The collections comprising these two genetic groups are geographically separated from each other and the other genetic groups by major roadways (Figs 1 and 3). This PCoA analysis of standardized genetic distances mirrors the geographic distribution of these genetic groups (Fig 1), suggesting a relationship between *R. norvegicus* genetic and geographic distance in this study system.

Fairly high estimates of global $F_{ST}$ were observed in the FSTAT analysis, with $\theta=0.184$ (95% CI = 0.155, 0.214). Of 66 pairwise $F_{ST}$ estimates among 12 *R. norvegicus* genetic groups, all were significant at $a=0.05$ and only 16/66 were not significant at the Bonferroni corrected $a=0.000758$. That said, pairwise $F_{ST}$ values were high among the 66 pairwise comparisons, ranging from 0.0826-0.2922 (S4 Table). All significant *p*-values (16/16) were associated with the Purple, Red, Orange, and Brown genetic groups, a result in agreement with the pairwise analysis of similarities (ANOSIM) (see below). Likewise, an Analysis of molecular variance (AMOVA) generated a moderate yet significant PhiPT value of 0.312 (p<0.001), revealing that ~31% of the molecular variance in this dataset was present among *R. norvegicus* genetic groups (*i.e.,* intersite variation), whereas 69% was present within them (*i.e.,* intrasite variation).

**Population genetic analyses: Isolation by distance and fine-scale genetic structure.** To explore signatures of isolation by distance, we used the RELATE function in PRIMER v5.2.9 and the standardized genetic distances and geographic locations (latitude and longitude) of the 247 individual *R. norvegicus* forming the 12 genetic groups (see Materials and Methods). The Spearman's rank correlation coefficient (Rho) was calculated as 0.232 (p<0.001), identifying a statistically significant relationship between *R. norvegicus* genetic distance and geographic distance. The global ANOSIM R statistic was 0.921 (p<0.0001), documenting that some of the 12 genetic groups formed distinct geographic groups. Most of the pairwise ANOSIM comparisons among the 12 genetic groups (61 of 66) were significant even with a very conservative Bonferroni corrected $a=0.000758$, indicating that most of the 12 genetic groups formed distinct geographic groups. Only 5/66 pairwise comparisons were not significant at $a=0.000758$ (Brown-Orange, Orange-Purple,

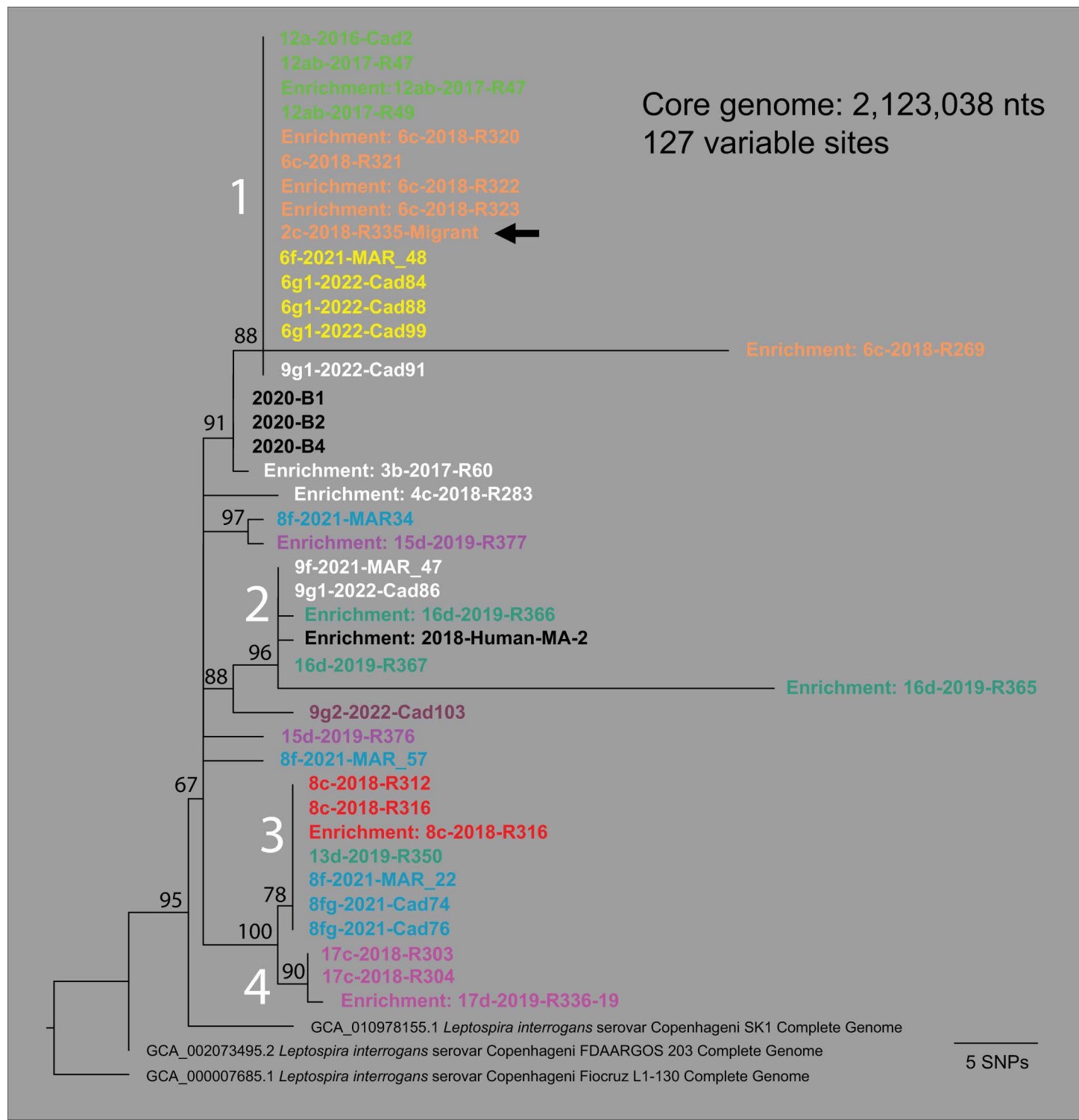

**Fig 2. *Leptospira interrogans* serogroup Icterohaemorrhagiae whole genome phylogeny.** A maximum likelihood phylogeny of 28 *Leptospira interrogans* serogroup Icterohaemorrhagiae isolate genomes from rats, two publicly available *L. interrogans* serogroup Icterohaemorrhagiae serovar Copenhageni complete genomes (GenBank accession# GCA_002073495.2 and GCA_010978155.1), and 13 enriched genomes (12 rats and 1 human) based upon a concatenated SNP alignment of 127 positions out of a core genome size of 2,123,028 nucleotides (nts). The phylogeny was

rooted with reference genome *L. interrogans* serogroup Icterohaemorrhagiae serovar Copenhageni strain Fiocruz L1-130 (GenBank accession# GCA_000007685.1). Four major clades were observed (1-4 in white text) and bootstrap values are indicated at specific nodes. The color of the genome ID corresponds to the rat genetic groups in Fig 1; black text indicates that no genetic group was assigned. One inferred migrant (indicated with a black arrow) from site 6c was trapped at site 2c and carried *Leptospira interrogans* identical to its source collection (see R335-Migrant in clade 1).

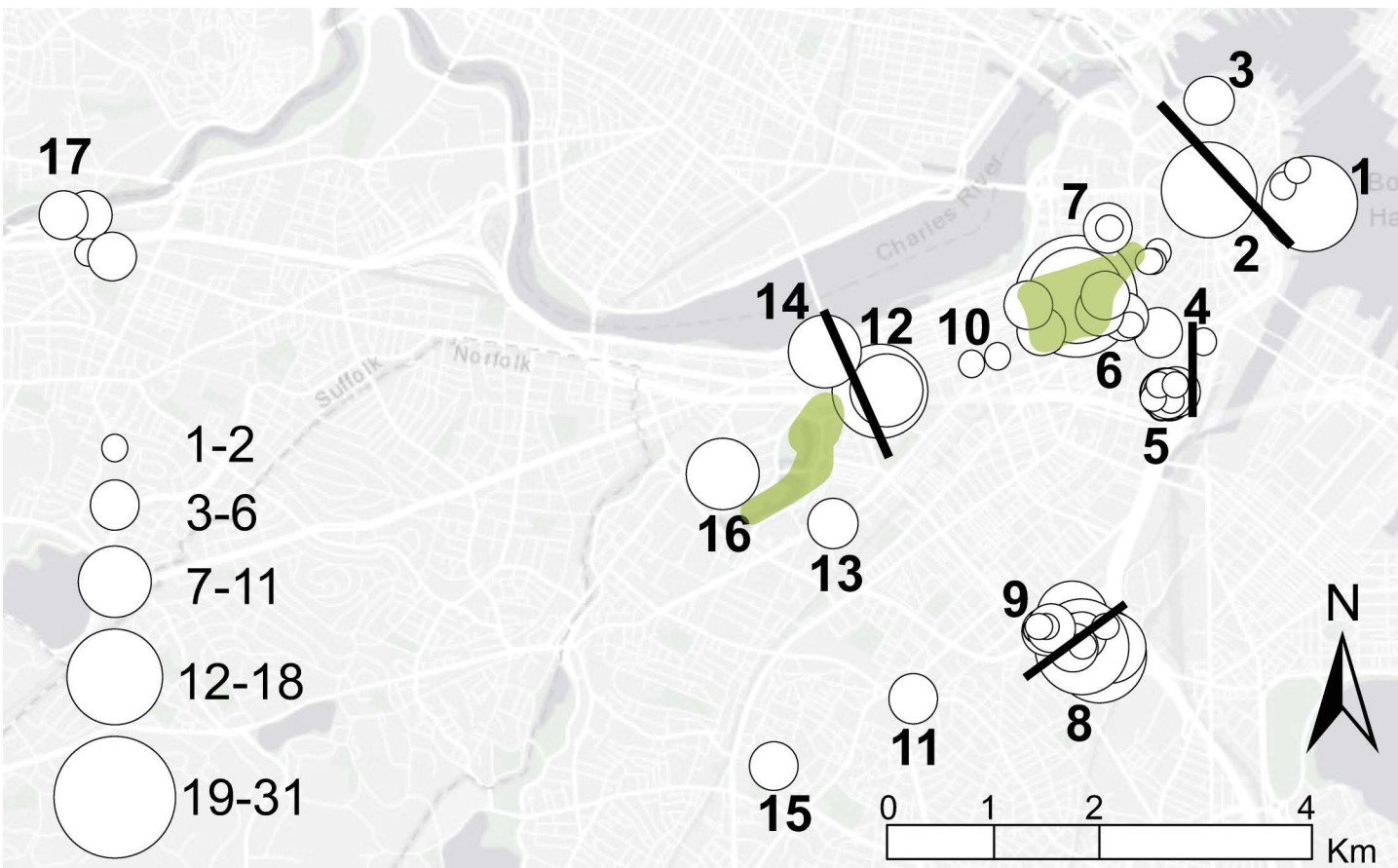

**Fig 3. *Rattus norvegicus* trap locations.** Map indicating 64 trap locations (white circles) at 17 sites (numbers), likely dispersal barriers (black bars), and major open spaces (green shading). Circle size corresponds to the number of total rat samples collected from each trap location. This map was created using ArcGIS software by Esri. ArcGIS and Arc-Map are the intellectual property of Esri and are used herein under license. Copyright Esri. All rights reserved. For more information about Esri software, please visit www.esri.com. Basemap: Light Gray Canvas Base https://www.arcgis.com/home/item.html?id=8b3d38c0819547faa83f7b7aca80bd76.

Purple-Red, Pink-Purple, and Purple-Evergreen), suggesting the Purple group has a genetic connection with three other groups (Red, Pink, and Evergreen).

Consistent with the signal of *R. norvegicus* genetic isolation by distance across all sites, microsatellite data revealed clear evidence of fine-scale spatial genetic structure within collection sites. First, the spatial autocorrelation analysis yielded correlation values significantly greater than zero in the first six distance classes (0–100 m out to 0–600 m), suggesting that, on average, relatives are generally living in close proximity (S4 Fig). This is supported by the significantly high value of average relatedness of individuals within collections, at 0.313 (95% CI = 0.268, 0.357), which indicates relatedness ranging from half (0.25) to full (0.5) siblings (S4 Table). Therefore, dispersing *R. norvegicus* would need to travel

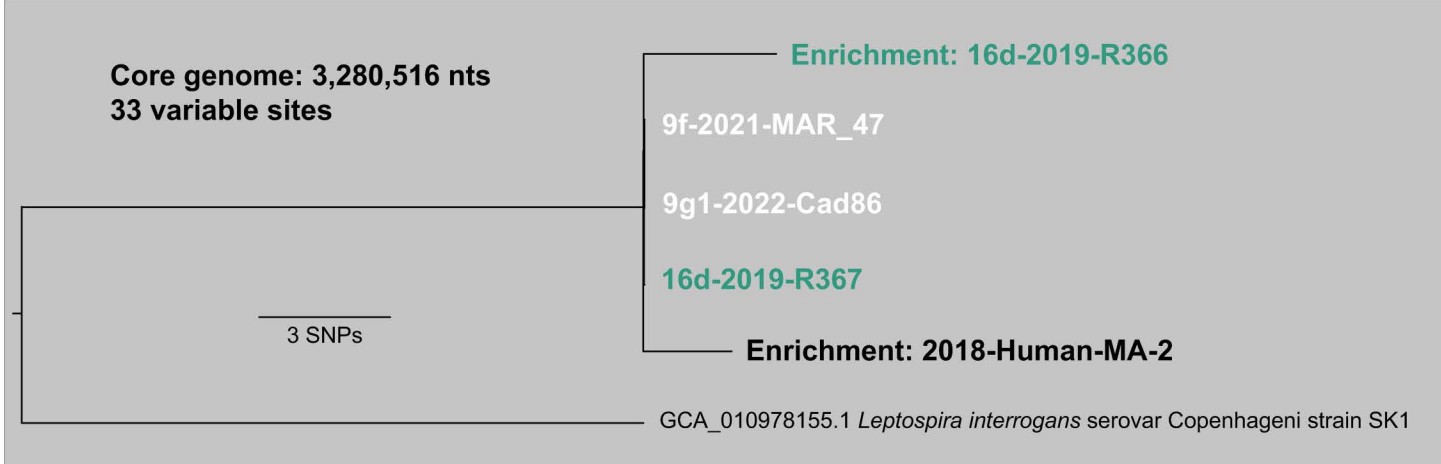

**Fig 4.** *Leptospira interrogans* **serogroup Icterohaemorrhagiae clade 2 whole genome phylogeny.** Maximum likelihood phylogeny of four *R. norvegicus* and one human derived *L. interrogans* genomes from clade 2 (see Fig 2) based upon the concatenated SNP alignment of 33 positions out of a core genome of 3,280,516 nucleotides. The color of the genome ID corresponds to the rat genetic group in Fig 1; black text indicates that no genetic group was assigned.

> 600 m, on average, to move beyond sites occupied by relatives. Finally, $F_{IS}$ values were only positive in three genetic groups [Blue (0.11), Yellow (0.078), and Red (0.006)], suggesting inbreeding within collections at sites 1 and 2 (Blue), and perhaps site 6 (Yellow) (Fig 1 and S4 Table).

### *Leptospira* spp. genomics and genotyping

In total, 52 of 59 *lipL32* PCR positive *R. norvegicus* and one of two positive human samples were assigned to *L. interrogans* (86.9%) using the genotyping methods described herein (S2 Table). Whole genome analyses revealed all isolates derived from frozen or fresh kidneys (n = 28) and all successful enrichments (n = 15) were *L. interrogans* (S2 Table) and most like other serogroup Icterohaemorrhagiae serovar Copenhageni/Icterohaemorrhagiae genomes (S5 Fig). Enriched genomes for samples R47 and R316, which were included as genomic analysis controls, were identical to their isolate genome counterparts (Fig 2). Phylogenomic analysis of all 28 isolates and 12 enrichments [the latter selected due to average breadth and depth of coverage values > 63% and > 18x, respectively (S5 Table)], revealed four distinct clades that largely associate with the genetic groups of the *R. norvegicus* from which they were derived (Fig 2). For example, 14 of 15 *L. interrogans* genomes assigned to clade 1 were obtained from rats belonging to the Yellow, Tan, and Green genetic groups, all occurring in or near Boston Common Garden, and the Tan and Yellow groups co-occur at site 6. However, the Tan group, the most prominent in Boston Common Garden in 2018, was replaced or modified as indicated by the emergence of the Yellow group in 2021 and 2022. Likewise, clade 3 contains mostly *L. interrogans* genomes from the Red and Steel genetic groups (6 of 7). Both genetic groups contain rats from the same site [8] but different collections and years (8b and 8c = Red, 8f and 8fg = Steel), perhaps indicating the removal of one rat genetic group (Red) and replacement with another (Steel). Similarly, clade 2 contains mostly *L. interrogans* genomes from rats belonging to the Evergreen genetic group (3 of 5); this clade also contains the *L. interrogans* genome derived from a human leptospirosis case in 2018 (MA-2). Finally, clade 4 contains only *L. interrogans* genomes from rats belonging to the Pink genetic group, comprised of the most geographically isolated collections (17c and 17d) (Figs 1 and S1). Our phylogenetic placement of two lower coverage enriched genomes (R289 and R292; see S5 Table) placed them into clade 1 among other *L. interrogans* genomes derived from rats from collection 6c and belonging to the Tan genetic group (S6 Fig).

Our higher resolution [3,280,516 nucleotides (nts)] phylogenomic comparison of *L. interrogans* clade 2 revealed the human derived genome (MA-2) from a 2018 leptospirosis case differed by only two single nucleotide polymorphisms (SNPs) from three *L. interrogans* genomes obtained from rats collected in 2019, 2021, and 2022, respectively, when comparing a core genome of > 3.2 mb (Fig 4). The four *L. interrogans* genomes comprising this clade were from rats belonging to the Evergreen genetic group or admixed group and were collected at sites 9 (collections 9f and 9g1) and 16 (collection 16d) (Fig 1 and S1 Table).

Our phylogenomic analysis of 24 whole genome assemblies representing all 18 P1 clade pathogenic *Leptospira* spp. placed the 24 assemblies into 18 unique species clades, as expected (S7A Fig). All nine isolate genomic DNAs subjected to AmpSeq analysis were accurately placed into their respective species clades (S7B Fig), which were conserved in the AmpSeq phylogeny despite the large reduction in size of the SNP alignment used to infer it (WGS = 53,132 vs. AmpSeq = 375 nts) and the core genome included in the analyses (WGS = 167,849 vs. AmpSeq = 1,195 nts) (S7 Fig). We note the proportion of variable sites (variable sites/core genome) represented in both the whole genome and AmpSeq phylogenies are essentially equal (WGS = 0.317 vs. AmpSeq = 0.314), which likely accounts for the similar branch lengths separating the species clades. The average number of reads that mapped for each isolate was 44,994 ± 12,348 and the number of AmpSeq loci that amplified for each isolate ranged from 22 for *L. kmetyi* to 39 each for *L. interrogans* and *L. santarosai* (S6 Table).

Of the 20 *R. norvegicus* and one human sample subjected to AmpSeq analysis, 13 rats yielded adequate PCR amplification and sequencing coverage (≥ 7 loci with > 10x reads) of *Leptospira* loci to confidently assign a species classification of *L. interrogans* (S6 Table). Nine displayed enough coverage (≥ 25 loci with > 10x reads) to assign within species phylogenetic placement to the *L. interrogans* serogroup Icterohaemorrhagiae serovar Copenhageni/Icterohaemorrhagiae clade (S8 Fig). The remaining seven rats and one human sample were not assigned a species classification due to inadequate PCR amplification and/or sequencing coverage (≤ 2 loci with > 10x reads); two rats (R66 and R108) had no mapped loci with > 10x reads, two (R324 and R379) had one mapped locus, and four samples (Cad1, MAR_36, MAR_5 and MA-1) had two mapped loci (S6 Table). Regardless, the few loci that did amplify for samples R324, R379, Cad1, MAR_36, MAR_5, and MA-1 all mapped to *L. interrogans* references except for MAR_5 and Cad1, which had one and two loci, respectively that mapped to *L. borgpetersenii*. The *L. interrogans* serovar Copenhageni strain Fiocruz L1-130 gDNA control amplified robustly at all 42 loci (1,444,384 assigned reads in total) and was accurately phylogenetically placed among other *L. interrogans* serovar Copenhageni strains (S8 Fig).

## Discussion

Our population genetic analyses of *R. norvegicus* in Boston revealed the existence of distinct rat populations for periods of up to five years, most likely due to low dispersal rates among rat populations and inbreeding. Our analyses revealed significant relatedness among individuals within a genetic group and, furthermore, that related individuals are generally living within close proximity to each other (S4 Fig), an observation consistent with other studies of *R. norvegicus* home ranges, dispersal, and site fidelity in urban settings [6,7,14,17,32–35]. In an analysis of gene flow among *R. norvegicus* in two US, one Canadian, and one Brazilian city, Combs et al., found that rats were highly related to each other within 500 m [12]. Our findings in Boston were similar, as rats would need to travel, on average, > 600 m to encounter an unrelated individual. Although inferred dispersal events were observed in this study (Fig 1, and S1 and S3 Tables), they were rare [only 11/311 rats (3.5%) were presumptive migrants], and this low rate is consistent with the observed patterns of strong genetic structure and stability of genetic groups over time (Fig 1).

The 11 instances of inferred migrations included seven trapped adults (5 females, 2 males) and four juveniles (1 female, 3 males) (S1 and S3 Tables). Distances ranged from ~0.8-5.3 km in Boston, but because of the methodology we used to assign rats to specific collections for population genetic analyses (see Materials and Methods), short-distance migrations (< 0.5 km) would only be detected among adjacent sites separated by major roads (*e.g.,* sites 8 and 9 in Fig

3). Although short-distance migrations are thought to be more common [6,7,36], long-distance migrations of up to 11.5 km have been documented in brown rats within urban settings [14] and could be anthropogenically facilitated [6,37]. In general, our observed rate of inferred migrations (3.5%) was similar to other studies of *R. norvegicus* population genetics in urban ecosystems; Gardner-Santana et al., reported a migration rate of 4.7% in Baltimore, US [14], and Kajdacsi et al., reported 6.8% in Salvador, Brazil [15].

Major roads were likely the largest barrier to *R. norvegicus* admixture and dispersal among neighboring sites in Boston (Figs 1 and 3). We observed several examples of major roads acting (and smaller roads failing to act) as dispersal barriers among three intensely sampled sites [6,8,9]. We compared the rat population dynamics within Boston Common Garden (site 6: sampled 2016–2022) that spans ~0.83 km and is bisected by a 3-lane road with those between Melnea Cass Blvd SE (site 8: sampled 2017–2022) and Melnea Cass Blvd N (site 9: sampled 2021–2022), which are directly adjacent to each other but separated by an 8-lane road (Figs 1 and 3). The 3-lane road bisecting the northeastern and southwestern portions of Boston Common Garden did not act as a dispersal barrier for *R. norvegicus* as both the Tan and Yellow groups were detected ubiquitously throughout the park. In contrast, the 8-lane road separating sites 8 and 9 apparently prevented admixing among them (Fig 1), despite the sites being separated by only ~0.25 km. Roadways have frequently been described as significant barriers to rat dispersal [6,7,17,35,38,39] and larger roadways deter dispersal more than smaller ones [40]. We also observed a pattern of genetic connectivity among sites separated by > 1 km (see Evergreen group; sites 13, 14, and 16 in Figs 1 and S1D) but connected by a large park complex containing green spaces, open water, and a network of river underpasses (Figs 1 and 3). Taken together, these results suggest different dispersal dynamics in commercial/residential areas (shorter movements) vs. park settings (longer movements) in this urban ecosystem, likely associated with road size, access to dispersal throughways, and resource availability. These observations are in concert with previous findings of *R. norvegicus* movement patterns in urban ecosystems [6,12,13,41].

*Rattus norvegicus* population turnover occurred at two sites [6,8], followed by robust expansion of the new/invading population (Figs 1 and S1). In Boston Common Garden we observed a large population in 2018 (collection 6c: Tan group) that was mostly displaced by the emergence of the Yellow group in 2021 (collection 6f). Both genetic groups remained (6g1 and 6g2) at the conclusion of sampling in 2022 although the Yellow group was most often detected in 2021 and 2022 (Fig 1B). We also observed genetic signatures of admixture among the Tan and Yellow genetic groups (individuals assigning to both groups or neither) in the later collections (6f and 6g2 in Fig 1B), which could suggest these genetic groups are in the process of merging. Complete population turnover occurred at site 8 between October 2018 (Red group) and June 2021 (Steel group) (Figs 1 and S1). A rat abatement campaign took place at site 8 but did not begin until 5-Nov-2021, suggesting population turnover from the Red to Steel group occurred before that campaign began. Furthermore, the Steel group persisted throughout the campaign (collections 8fg and 8g in Figs 1 and S1). It is unknown what occurred at site 8 between October 2018 and June 2021 that led to the population change, but our results suggest that rat abatement is unlikely to lead to population turnover unless complete eradication can be achieved, as populations can rapidly rebound to pre-intervention sizes once interventions cease [42–45]. However, if an intervention resulted in removal of dominant individuals followed by immigration of other individuals into this open niche, population turnover could occur [7,42]. We suspect an open niche once recolonized would result in the rapid expansion of a new genetic group due to rapid urban rat reproduction rates (up to five litters per year) [7] paired with the tendency of *R. norvegicus* to breed with individuals from their same genetic group, a result of dispersal barriers, site fidelity, and small home ranges [6,12].

*Leptospira interrogans* persists locally in *R. norvegicus* populations in Boston. We detected pathogenic *Leptospira* spp. DNA in 18.0% of rats [including 2 of 11 (18.2%) inferred migrants] representing 20/33 collections and 12/17 sampling sites (Figs 1 and S1, and S1 Table) with 98.3% of infected rats classified as adults. To put these number into context, several other studies that examined *Leptospira* spp. carriage in *R. norvegicus* in urban ecosystems reported 7–82% in the highly endemic tropical climate city of Salvador, Brazil [46–48], 11.1% in the temperate city of Vancouver, Canada [49], and 35.7% in New Orleans, US [50]. Furthermore, a 2019 review of *Leptospira* spp.

carriage in rats globally reported an overall carriage rate in *R. norvegicus* of 30.3% [25], with higher carriage rates associated with tropical climates. Of the studies that assessed the relationship between *Leptospira* spp. positivity and age or sexual maturity, all reported significantly higher proportions of positive adults than juveniles [47–49]. Using a suite of molecular methods, we determined the *Leptospira* spp. circulating in *R. norvegicus* in Boston was *L. interrogans* and most similar to other serogroup Icterohaemorrhagiae genomes (S5 and S8 Figs). The 28 cultured isolates were confirmed to belong to serogroup Icterohaemorrhagiae via the microscopic agglutination test (MAT) (S2 Table), whereas the samples subjected to enrichment (n = 14) and AmpSeq (n = 9) could only be inferred based upon phylogenetic placement (Figs 2, S5, S6, and S8). Detection of only *L. interrogans* serogroup Icterohaemorrhagiae in this rat dataset was not surprising. Indeed, the exclusive detection of *L. interrogans* serogroup Icterohaemorrhagiae in *R. norvegicus* from the US was documented in a recent review [25]. In that same article, it was also reported that the most frequently identified *Leptospira* spp. in rats globally (in 36 of 43 countries) belong to serogroup Icterohaemorrhagiae. To that point, in the broad phylogenetic framework of *L. interrogans,* the genomes from Boston *R. norvegicus* formed a monophyletic clade with four other *L. interrogans* serogroup Icterohaemorrhagiae genomes from Saint Kitts (SK1), Brazil (203 and L1-130), and Malaysia (898) (S5 Fig); this analysis included diverse *L. interrogans* representing > 9 serovars. Importantly, monophyletic groupings like this can be teased apart with clade-focused whole genome analysis, which was demonstrated by Santos et al., in a comprehensive study of 67 *L. interrogans* serogroup Icterohaemorrhagiae isolates originating from >15 countries across the globe [51]. At that scale, they describe the detection of 1,072 SNPs and 258 indels among the 67 isolates, and phylogenetic clusters that broadly associated with geographic locations. Likewise, by using clade-focused whole genome analysis we identified > 100 SNPs and four distinct clades among *L. interrogans* serogroup Icterohaemorrhagiae genomes from *R. norvegicus* in Boston (Figs 2 and S6).

Distinct clades of *L. interrogans* correspond to specific geographic locations and rat populations in Boston and persist locally through time despite population turnover (Fig 2). Rats from sites 6 and 12 exclusively carried *L. interrogans* from clade 1 (Figs 2 and S6), the virulence of which was validated in the hamster model of leptospirosis, despite being separated by > 1.3 km and belonging to three different rat genetic groups (Tan, Yellow, and Green); at site 6 the rat population shifted from the Tan to Yellow group between 2018 and 2021 (Figs 1 and S1). Furthermore, collections from sites 6 and 12 spanned seven years (2016–2022) (S1 Table). This finding suggests connectivity among these sites and/or that stable rat populations and favorable environmental conditions at these sites maintain endemic clades of *L. interrogans* over time. A greenbelt containing a public walkway connects these two sites almost directly, and although eight roads bisect this greenbelt, none exceed 4-lanes and six are only 2-lanes, suggesting rat movement between these sites is possible [40]. Likewise, *L. interrogans* from clade 2 was exclusively found in rats from sites 9 and 16 (Figs 2 and S6). These sites were separated by nearly 2.5 km and the samples were collected over different years; the site 16 rats were trapped in 2019 and the site 9 rats in 2021–2022. As with clade 1, we see two distant locations sharing a single *L. interrogans* genotype, suggesting a connection between them, but in this case there is no obvious dispersal throughway linking them. That said, we see evidence of long-distance migrations throughout this study system (Fig 1), so it is plausible that a *Leptospira*-positive migrating rat from site 16 to site 9 (during or post 2019) led to the establishment of the clade 2 genotype at site 9 that we observed in 2021–2022. Population turnover of the rats with persistence of the same *Leptospira* spp. genotype was also observed at site 8 (Figs 2 and S6) where the rats displayed a genetic change from Red to Steel between 2018 and 2021, yet the *L. interrogans* clade 3 genotype remained the same. Finally, *L. interrogans* clade 4 contained only rats from site 17 collected in 2018–2019. Site 17 is the most geographically distant site in this study being > 4.5 km away from its nearest geographic neighbor, site 16 (Figs 1, 3 and S1), so it is not surprising to see an exclusive clade of *L. interrogans* among rats from site 17. These results provide a clearer picture of *Leptospira* spp. transmission among rat populations and provide compelling evidence for local persistence of distinct *L. interrogans* serogroup Icterohaemorrhagiae genotypes among *R. norvegicus* in Boston.

*Leptospira* spp. genotypes could be dispersed between populations by migrating rats. We detected the likely migration of a *Leptospira*-positive adult rat (R335) from site 6 to site 2 in 2018 (Fig 1) and the genetic signature of the *Leptospira* spp. it carried (clade 1) further supported this assignment (Fig 2). Importantly, this event illustrates a mechanism by which *Leptospira* spp. genotypes could be transferred between rat populations. We also observed single rat populations that carried diverse *L. interrogans* that assigned to multiple locations in the phylogeny (*e.g.*, 15d, 9g1, 8f) (Fig 2), suggestive of transportation of *L. interrogans* between rat populations. For example, a single rat trapped at site 9 (Cad91) contained *L. interrogans* from clade 1 (Fig 2). Although this rat was not assigned to a genetic group, the most probable assignment was to the Brown group ($Q = 0.589$) (S3 Table). We see evidence of rat mediated *L. interrogans* transportation at site 8 where three *L. interrogans* genotypes were recovered (Fig 2). Because the clade 3 genotype was the most common at site 8 and maintained over three years despite the turnover of the rat population from the Red to Steel group, we suggest the clade 3 *L. interrogans* is established at that site, whereas the other two genotypes ($n = 1$ each) were the result of *Leptospira*-positive rat migrations into site 8. Finally, we found clade 3 *L. interrogans* in a rat from site 13 that clearly assigned to the Evergreen genetic group (Fig 1); clade 3 genotypes were otherwise only found in rats from site 8 (Fig 2). Taken together, these data support *R. norvegicus* migrations being one mechanism by which *L. interrogans* serogroup Icterohaemorrhagiae genotypes are transferred among rat populations in Boston, an important consideration when designing rodent control interventions or construction, as both can drive long-distance migrations and intraspecific antagonism [42] and human interventions may increase rat population pathogen prevalence [52].

The mechanism(s) by which identical *Leptospira* spp. genomes become established in rat populations at these sites is unclear. *Rattus norvegicus* are well known chronic and asymptomatic carriers of *L. interrogans* serogroup Icterohaemorrhagiae [20–22], and the transmission cycle via carriage of *Leptospira* spp. in renal tubules, excretion through urine, and subsequent environmental contamination ultimately leading to direct or indirect exposure to other rats as well as other susceptible mammals, including humans, is well established [24]. During chronic carriage in rats, one might expect to observe the accumulation of mutations over time corresponding to an inherent *L. interrogans* mutation rate (to our knowledge, this is unknown). Indeed, we do observe this pattern across certain rats (R269, R289, R292 in clade 1; R365, R366 in clade 2; R336-19 in clade 4) (Figs 2 and S6), presumably the result of long-term chronic infection during which *L. interrogans* was in a replicative state. More often, however, we observed identical genomes from rats spanning up to a seven-year period, suggesting a dormant or very low replication state of *L. interrogans* in this urban ecosystem. A possible mechanism for this could be biofilm formation [53] in the renal tubules of the rats [54]. However, laboratory analysis of *R. norvegicus* infected with *L. interrogans* serovar Copenhageni revealed they shed leptospires continuously ($10^5$ -$10^7$ leptospires per mL) in their urine for the duration of the experiment (159 days) [22], suggesting that in this laboratory setting there was no such dormant state. Alternatively, *Leptospira* spp. biofilm formation could take place in the environment [55,56]. It is important to note that because we are only analyzing the shared genomic space among all the rat-derived *L. interrogans* genomes (core genome > 2.1 million nts) there could be mutations or acquired/lost pan genomic content outside of the core genome that we did not detect. That said, if replication were occurring at a constant rate, we would still expect to observe mutations accumulating in the core-genome. Additionally, when we increased our resolution for clade 2 to > 3.2 million nts (Fig 4), we observed the same identical genomes shared among three rats (MAR-47, Cad86, R367). Furthermore, our genomic analysis of clade 2 also included a human derived *L. interrogans* genome (MA-2 in Figs 2 and 4). This genome was enriched from urine DNA obtained from a human leptospirosis case in 2018 and only differs by two SNPS (core genome > 3.2 million nts) from *L. interrogans* genomes obtained from three rats at sites 9 and 16 in 2019 and 2022 (Fig 4). Due to the clonality of these *L. interrogans* genomes derived from two host species [human (incidental) and rat (reservoir)] collected over a five-year period (2018–2022) paired with the knowledge that rats are recognized globally as asymptomatic and chronic carriers of serogroup Icterohaemorrhagiae [whereas other hosts (*e.g.*, dogs) are incidental carriers that would likely suffer acute disease], we suggest the most likely explanation for the human infection is exposure to a fomite or the environment that was contaminated by rats from sites 9 or 16 or another *R. norvegicus* population

carrying the same *L. interrogans* clade 2 genotype in the city. That said, any attempts to link this human case to a specific site should be interpreted with caution given that *L. interrogans* dispersal between sites likely occurs (Figs 1 and 2).

This is the first study to pair fine-scale population genetic analyses of *R. norvegicus* in an urban ecosystem with genomic analysis of the *L. interrogans* they harbor. This analysis was possible due to improvements in culturing techniques (described herein) enabling high rates of *Leptospira* spp. recovery (47.5%) paired with advances in culture independent molecular capabilities [28] that together enabled the acquisition of 40 *L. interrogans* genomes from rats (out of 59 *lipL32* PCR positives) and one from a human. Importantly, these genomes were derived not just from fresh samples, but also frozen and archived samples (tissues and DNA), thereby greatly expanding the types of samples that could be included in these genomic analyses. A particularly salient outcome was the acquisition of a human derived *L. interrogans* genome from trace amounts of archived DNA (22.4 ng total). Although not much is known about the origin of this urine sample beyond it originating from a human leptospirosis case in Boston in 2018 due to patient privacy rights, our analysis placed it among rat derived genomes collected exclusively at sites 9 and 16 (Fig 4), suggesting the source of infection was localized to a relatively small geographic area within Boston (Fig 1) that was accessible to rats from both sites.

We have previously applied AmpSeq to assess the genetic diversity of *lipL32* sequences from pathogenic leptospires [original design described by Stone et al., [56]]. AmpSeq is based upon traditional PCR methods and Illumina short-read sequencing (both technologies that are accessible to researchers and clinicians across the world) and requires < 10 μL of a *Leptospira*-positive clinical DNA sample. Furthermore, it is relatively inexpensive on a per-sample basis due to the ability to simultaneously sequence hundreds of samples on a single Illumina run. For this study, we implemented and assessed an expanded *Leptospira* spp. AmpSeq panel that can provide *Leptospira* spp. identification among all 69 currently described and validated *Leptospira* spp. from the pathogenic P1 and P2 clades and the saprophytic S1 and S2 clades [57] (S1 File), as well as sub-species phylogenetic placement within *L. interrogans* (*i.e.*, serogroup associated clades) (S8 Fig). Although we did not extensively assess sub-species placement among other P1 *Leptospira* spp., we expect a similar outcome for *L. kirschneri*, *L. noguchii*, *L. alexanderi*, *L. santarosai*, *L. weilii*, *L. borgpetersenii*, and *L. mayottensis* due to the high number of discriminating loci (< 35) expected to amplify in each species (S1 File). Furthermore, our phylogenetic analysis of P1 clade pathogenic *Leptospira* spp. via sequences generated with AmpSeq suggests sub-species placement (S7B Fig), even with as few as 22 amplicons (S6 Table). Importantly, the AmpSeq method can identify mixtures of *Leptospira* spp. within a single sample [56], which is an under-characterized aspect of *Leptospira* spp. ecology. Molecular advances such as these will greatly facilitate an increased understanding of mixed infections and their potential role in leptospirosis infection and disease ecology. Thus, we suggest it could serve as an applicable tool for species and sub-species identification of circulating and clinically relevant leptospires in regions throughout the world.

## Conclusion

In this study, we paired population genetic analyses of *R. norvegicus* in an urban setting with genomic analyses of the infecting leptospires they transmit to gain a deeper understanding of the nuanced transmission patterns of *Leptospira* spp. among rats and spillover into humans. We found that rats in Boston displayed strong signals of population structure, likely associated with behavior and dispersal barriers. We also found the *Leptospira* spp. carried by rats in Boston was *L. interrogans* and most similar to other serogroup Icterohaemorrhagiae isolates, and that there was fine-scale genetic structure that associated with distinct rat populations. Our analysis identified highly clonal genomes of *L. interrogans* that persisted within certain populations of rats for up to seven years. Finally, our genomic analysis of *L. interrogans* obtained from a human leptospirosis case using DNA capture and enrichment strongly suggests a link to rats as the source and provides a likely geographic origin within Boston where this transmission event might have occurred, although we cannot rule out the possibility that the same genotype of *L. interrogans* was also carried by rats in an unsampled location in Boston.

## Materials and methods

### Ethics statement

Ethical approval for rodent trapping was granted by the Cummings School of Veterinary Medicine Institutional Animal Care and Use Committee. Ethical approval to evaluate virulence using a hamster model was granted by the Animal Care and Use Committee at the National Animal Disease Center, United States Department of Agriculture.

### Study location

The City of Boston, Massachusetts (125.4 sq. km), in the eastern US has an estimated population of ~650,000 people (US Census Bureau 2020 Decennial Census) with a temperate climate (cold winters and hot, humid summers). *Rattus norvegicus* were trapped over a seven year sampling campaign (2016–2022) within 17 geographic sites in Boston (Fig 3). Each site was the approximate size of a few city blocks (see site definitions below). Three sites were sampled intensely between August 2021 and August 2022: Boston Common Garden (site 6), Melnea Cass Blvd SE (site 8), and Melnea Cass Blvd N (site 9) (Fig 3). These three sites were identified in consultation with the COB-ISD as having active rat populations and burrow systems. They were selected for intensive sampling to facilitate a deeper understanding of the relationship between rat population genetics and leptospirosis transmission in parks versus mixed commercial/residential environments, and to assess how a major road may impact dispersal among neighboring rat populations, which we hypothesize will impact the transportation of pathogenic *Leptospira*. Furthermore, we used these sites to assess how genetic signatures change over time across stable rat populations (sites 6 and 9) and the effects of a rat control intervention (baiting and trapping) at site 8 that began 5 November 2021. Boston Common Garden is a heavily landscaped urban park system with primarily mowed grass ground cover and concrete foot paths. The southwest and northeast portions are bisected by a 3-lane road, and the southwest portion is the site of a historical municipal burial ground. The two Melnea Cass Blvd sites are heavily developed urban areas of mixed commercial and residential use and separated by an 8-lane road.

### Rodent trapping

*Rattus norvegicus* sampling between July 2016 and November 2020 was conducted in collaboration with the City of Boston's Inspectional Services Department (COB-ISD), as described previously [10,11]. For the three sites that were sampled intensely between August 2021 and August 2022, 6–10 traps were placed at each site at least 15 meters apart in locations with evidence of recent rat activity (*i.e.*, near burrows or along runs with evidence of fresh rat droppings/urine). The traps consisted of a commercial lethal snap trap (T-Rex, Bell Labs, Windsor, WI, US) fixed to a remote sensing device (Arctic Systems, Copenhagen, DK) with zip-ties placed inside a weighted, locked commercial bait station (Protecta EVO Express Bait Station, Bell Labs, Windsor, WI, US). Traps were baited with meat jerky (Slim-Jim original flavor, Conagra Brand, Chicago, IL, US) and set and baited in the evening, attended from dusk until dawn, and deactivated and shut down the following morning. The remote sensing device alerted the study team in real time for prompt retrieval of carcasses and reset of the trap. We attempted to trap rats every 1–4 months, but actual trapping effort varied and was determined by convenience, and thus not included as a metric in the analysis.

Carcasses were stored in a cooler with icepacks until necropsies were performed within six hours of trapping. Carcasses were visually identified as brown or Norway rats (*Rattus norvegicus)* based on their external appearance and the morphology of their ear and tail [10,11]. Rats were weighed, sexed, dipped in 70% ethanol, and placed in dorsal recumbency. Sterile tissue scissors and forceps were used to reflect the abdominal skin and a new set of sterile instruments were used to open the abdomen and aseptically harvest the kidneys. For samples collected in 2021–2022, one fresh kidney was placed into 50 mL conical tubes containing 10 mL liquid HAN medium [26] and shipped overnight at ambient temperature to the National Centers for Animal Health (NCAH) in Ames, IA. Archived frozen kidneys (2016–2020) were stored at -80°C until they were shipped overnight from Tufts University to NCAH on dry ice.

### *lipL32* qPCR detection of *Leptospira* in rodent samples

We screened for the presence of pathogenic *Leptospira* spp. DNA in DNA extracted from 210 frozen kidneys, 118 fresh kidneys, and one fresh urine sample (kidney was also evaluated from this individual) from 328 *R. norvegicus* rats collected in Boston from 2016-2022; four kidneys from bycatch mice and one kidney from a gray squirrel were also included. Kidneys were transferred to a 150 mm petri dish, the capsule (fibrous outer layer of the kidney) removed, and then placed in a 710 mL Whirl-Pak bag (World Bioproducts, LLC, Madison, Wisconsin, US) containing 10 mL of liquid HAN, and immediately macerated manually. DNA was extracted using 500 μL of kidney macerate or 200 μL of urine, and the Maxwell RSC Purefood Purification Pathogen kit (Promega Corporation, Madison, Wisconsin, US). We screened all samples for the presence of pathogenic *Leptospira* spp. using qPCR assay specific for the *lipL32* gene as previously described, using primer set LipL32-47Fd and LipL32-301Rd with LipL32-189P probe (Thermo Fischer Scientific, Waltham, MA, US; [58–60]). To control for qPCR inhibitors, we added an exogenous Internal Positive Control (Thermo Fischer Scientific, Waltham, MA, US) to the master mix to confirm DNA amplification, detect false negatives, and qualitatively detect presence of amplification inhibitory substances. All samples were assayed in triplicate and considered positive when duplicate or triplicates were positive with Ct values < 40 [58–60]. Using chi-square tests of independence, we examined associations between *Leptospira*-positive rats and six sampling variables: sex, maturity (juvenile vs. adult), sampling season [spring (19-Mar to 20-Jun), summer (20-Jun to 22-Sept), fall (22-Sept to 21-Dec), and winter (21-Dec to 19-Mar)], year, site, and rat genetic group (assigned herein).

### Culture

Remaining tissue from frozen and fresh kidneys positive by *lipL32* qPCR was used for culture. Macerate from frozen kidneys (200 μL) was inoculated into 5 mL of semisolid T80/40/LH media [61] and incubated at 29°C. Two 10-fold serial dilutions were made from the fresh kidney macerate into liquid HAN media in 3% $CO_2$ [26] and T80/40/LH media [61] and incubated at 37°C and 29°C, respectively. Cultures were maintained for up to six months and checked periodically by dark-field microscopy for the presence of leptospires.

### Serotyping of isolates

*Leptospira* spp. isolates were serotyped by the MAT with a panel of polyclonal rabbit reference antisera representing 13 serogroups: Australis, Autumnalis, Ballum, Bataviae, Canicola, Grippotyphosa, Hebdomadis, Icterohaemorrhagiae, Mini, Pomona, Pyrogenes, Sejroe, and Tarassovi. The serogroup of each isolate was assigned according to the reference antiserum yielding the highest agglutination titer.

### Evaluation of virulence

One rodent isolate (designated strain R47) was propagated in liquid HAN medium at 37°C in 3% $CO_2$ and evaluated for virulence by intraperitoneal injection of $10^8$ leptospires in a final volume of 1 mL into one group of four female golden Syrian hamsters (*Mesocricetus auratus*). Negative control hamsters were inoculated with media alone. Hamsters were monitored daily for clinical signs of disease, including weight loss, lethargy, bloody discharge from the nose or urogenital tract, and sudden death; and were euthanized when moribund. At time of euthanasia, liver and kidney tissue were harvested for culture and *lipL32* qPCR.

### *Rattus norvegicus* collection site definition

Throughout the entire study period (2016–2022), traps were place in 64 distinct locations (S7 Table) assigned to 17 geographic sites in Boston (Fig 3). Site numbers were assigned from east to west and defined as trapping grids separated by > 0.5 km (Fig 3). Exceptions to this were trapping locations ≤ 0.5 km apart but separated by major roadways, which

are known dispersal barriers for *R. norvegicus* and *R. rattus* [6]; these were considered separate sites. A large city park complex (~0.83 km northeast to southwest) comprised of two smaller parks [Boston Common (northeast) and Boston Public Garden (southwest)] was considered a single site and is referred to hereafter as Boston Common Garden. We grouped individual rats into 33 collections based upon site ID and sampling date, whereby samples collected > 4 months apart were considered unique collections. These criteria were selected based upon the ~3 months needed to reach sexual maturity for *R. norvegicus* [62] and previous work describing genetic relatedness among *R. norvegicus* in urban settings with distances of up to 0.5 km [12]. Thus, nomenclature for collections is as follows: site is indicated by numerals ranging from 1-17 ordered from east to west followed by letters corresponding to year of collection where a = 2016, b = 2017, c = 2018, d = 2019, e = 2020, f = 2021, and g = 2022. If multiple collections were obtained within a year, a second numeral follows the letter designation (*e.g.,* 9g2 indicates the second collection at site 9 in 2022). These definitions resulted in 33 collections from 17 sites (S1 Table).

### *Rattus norvegicus* microsatellite development and genotyping

**Marker selection, optimization, and multiplexing.** Twenty-one MSAT markers were utilized for population genetic analyses of *R. norvegicus* samples (Table 1). All markers have been previously published [14,15,17,32,67] and were selected for use in this study based upon: 1) geographic regions where they were previously utilized (*i.e.*, eastern US cities as a proxy for presumed genetic compatibility with our Boston study site), 2) number of described alleles in other study systems (prioritizing markers with multiple alleles), and 3) successful use among diverse populations of *R. norvegicus* as inferred by inclusion in multiple studies from diverse western hemisphere countries (*i.e.,* US and Brazil). The 21 markers were optimized using a panel of six *R. norvegicus* DNA samples from different sites in Boston, one Sprague Dawley rat gDNA control (Zyagen, San Diego, CA, US), and a non-template control (molecular grade water).

We generated multi-locus genotypes for 328 rat kidney DNAs using these 21 MSAT markers amplified in nine PCR reactions (six multiplexes and three singles, Table 1). All PCRs were carried out in 10 μL volumes containing the following reagents (given in final concentrations): 10–20 ng of DNA template, 1x PCR buffer, 2.5 mM MgCl2, 0.2 mM dNTPs, 0.8 U Platinum Taq polymerase (Invitrogen, Carlsbad, CA, US), and 0.1-0.5 μM of each primer pair (see Table 1). Thermocycling used the following conditions: 10 minutes at 95°C to release the Platinum Taq antibody, followed by 38 cycles of 60 s at 94°C, 30 s at the annealing temperature (Ta) and 30 s at 72°C. The Ta, dilution, multiplex, and pooling schemes for each marker are provided in Table 1. Diluted PCR products were electrophoresed on an ABI 3130xl sequencer with a dye-labeled high density size standard (Applied Biosystems, Foster City, CA, US) and analyzed using genotype analysis software (SoftGenetics, State College, PA, US).

**MSAT sample data filters.** Our final MSAT dataset included only rats with complete genotypes at ≥ 19 markers; one individual sample failed to meet this criterion and was removed. We also removed three individuals with evidence of DNA cross contamination as indicated by the presence of > 2 alleles at multiple markers. Finally, location provenance was incomplete for 13 individuals (including all eight rats collected during 2020), which were also removed. This resulted in a final MSAT dataset of 311 *R. norvegicus* representing 33 collections from 17 sites collected between 2016–2019 and 2021–2022 (S1 Table).

### MSAT marker validation analyses

We ran a series of tests on the 21 MSAT markers to validate neutrality and utility in population genetic analyses of *R. norvegicus* from Boston using all 311 individuals organized into 33 collections. Validations included a query for identical multi-locus genotypes in GenAlEx [68] and a test for linkage disequilibrium in FSTAT v2.9.4 based on the log-likelihood ratio G statistic [69]. We used MICRO-CHECKER [70] to test for null alleles, large allelic dropouts, and scoring errors due to PCR stuttering using the following settings: maximum expected allele size of 420 bp, a 95% confidence interval (CI), and 1000 repetitions. We performed a test of Hardy-Weinberg equilibrium for each marker, also in GenAlEx, but using

**Table 1.** *Rattus norvegicus* MSAT marker primers and conditions.

| Marker | $T_a$ | Mix | Primer pair conc. (µM) | Post-PCR dilution | Dye | *A* | Allele sizes (bp) | Citation and primer sequence (5'-3') |
|---|---|---|---|---|---|---|---|---|
| D12Rat76 | 60 | 1 | 0.5 | 1/50 | NED | 10 | 86, 88, 90, 92, 94, 96, 98, 100, 102, 104 | [1] F-TGCCTTTTAAAATGATGTGCA R-ATTGGCAATGCACTCATGTG |
| D1Wox23 | 60 | 1 | 0.3 | 1/50 | VIC | 10 | 186, 190, 194, 198, 202, 206, 210, 214, 218, 222 | [2] F-TCTGACCCATACTTGTACTTTGC R-AATTTCTGCCTCTTTTTCTCAG |
| D7Rat97 | 60 | 1 | 0.3 | 1/50 | 6FAM | 11 | 172, 174, 176, 178, 180, 184, 186, 188, 192, 196, 198 | [3] F-CAAGTTTTCCTCTGCCCAAG R-GCTGTCATTCCACTGGGTTT |
| D11Mgh5 | 60 | 2 | 0.4 | 1/75 | 6FAM | 9 | 216, 226, 228, 230, 232, 234, 236, 238, 248 | [1] F-CAGCTCTAATTCCAGAAAGGTTT R-GAATCGATTGACAGATGTCTGTG |
| D3Cebr3 | 60 | 2 | 0.3 | 1/75 | PET | 5 | 163, 165, 167, 177, 181 | [4] F-CAGGGAATGCAGAAGATACAG R-GTGGCTTTAGGACTCTGGAG |
| D6Cebr1 | 60 | 2 | 0.4 | 1/75 | NED | 7 | 220, 224, 226, 230, 232, 234, 236 | [4] F-TGGTTTGGTTGGGGAGAA R-GTGCTGTCAGGGAAAGATGTA |
| D4Wox7 | 55 | 3 | 0.1 | 1/50 | VIC | 5 | 130, 134, 138, 142, 146 | [2] F-GATAGCATAAAATCCCTAGAGGTT R-TCGATTTATCTGAAACCATCAC |
| D6Wox2 | 55 | 3 | 0.3 | 1/50 | 6FAM | 10 | 96, 100, 104, 108, 116, 120, 124, 128, 132, 136 | [2] F-CCAGTCCATACTTATCCATCTG R-CATTTAGATAGGTGATAGATTCAG |
| D1Cebr9 | 55 | 3 | 0.3 | 1/50 | 6FAM | 6 | 261, 263, 265, 267, 269, 271 | [4] F-GGATTTGGCTCCCTTTAAG R-CAGTAACTCTGGTTCATGTACTCC |
| D7Rat13 | 64 | 4 | 0.4 | 1/25 | NED | 12 | 128, 144, 146, 148, 150, 152, 154, 156, 158, 160, 162, 164 | [5] F-GACTTCTGCTACACGCCACA R-CAGCCCTAGAAGGAAATGCA |
| D5Rat83 | 64 | 4 | 0.4 | 1/25 | VIC | 11 | 164, 168, 170, 178, 184, 186, 190, 192, 194, 196, 200 | [5] F-ACTTGGAAACAGGGAGATGG R-GGGTCTTCAGGATGGCAATGT |
| D12Wox1 | 64 | 4* | 0.4 | 1/50 | 6FAM | 8 | 392, 396, 400, 404, 408, 412, 416, 420 | [2] F-GACATTAAGGGGTCTTCCTAAG R-TATCTTTGCAACGCTGAGG |
| D10Rat20 | 64 | 5 | 0.4 | 1/25 | PET | 6 | 109, 113, 115, 117, 119, 123 | [1] F-AGTGATTGCCATACCTGCCT R-GAAATGGCCAGGATAAACCA |
| D20Rat46 | 64 | 5 | 0.4 | 1/25 | 6FAM | 13 | 140, 142, 144, 148, 154, 156, 160, 162, 164, 168, 172, 174, 180 | [1] F-AAGTACTGAGTGGGCTGCGT R-GGCAAAACACCAATGCCTAT |
| D10Mit5 | 64 | 5 | 0.2 | 1/25 | NED | 9 | 135, 137, 139, 141, 143, 145, 147, 149, 151 | [1] F-TGCTGGGTGAACCAGAGAG R-CTGCCCTCCAAACCACC |
| D9Rat13 | 64 | 5 | 0.4 | 1/25 | VIC | 5 | 101, 109, 119, 121, 123 | [5] F-CCCATCTTTACACCTCCCAA R-GGAAAGGAAACTGGAGGGTC |
| D19Wox11 | 66 | 6 | 0.3 | 1/15 | NED | 5 | 212, 216, 220, 224, 228 | [2] F-CTACCCACCCATCTATTCATCC R-GTTTCCAGCACCCATGTCC |
| D1Cebr3 | 66 | 6 | 0.3 | 1/15 | VIC | 6 | 79, 81, 83, 105, 107, 109 | [4] F-CTTGGGAGCTGGGAGTGT R-GAAGGCTGAGGTATGAAGACTG |
| D16Rat81 | 66 | 6 | 0.4 | 1/15 | PET | 7 | 148, 150, 152, 154, 156, 158, 160 | [1] F-GAGCCTTAGCACAGTGGCTT R-GGCCCACATGTGCATGTATA |
| D5Rat33 | 66 | 6* | 0.4 | 1/15 | 6FAM | 15 | 102, 106, 110, 112, 116, 118, 120, 122, 124, 126, 130, 132, 134, 138, 140 | [3] F-TGGAGAAAAGAAGAACCTCCA R-GTGCCCTCAGACTGAACTC |
| D1Cebr4 | 66 | 6* | 0.4 | 1/30 | VIC | 8 | 278, 284, 286, 288, 290, 292, 294, 296 | [4] F-GACCTCCTGCCCCTTCACTG R-TGAAAAATGAATTGCTTGTG |

a [1] [63], [2] [64], [3] [65], [4] [66], [5] [67].

b Additional single PCRs are pooled into a corresponding mix (*i.e.*, multiplex PCR) prior to capillary electrophoresis. $T_a$ = PCR annealing temperature. *A* = number of observed alleles in Boston, MA *Rattus norvegicus* (2016–2019 and 2021–2022).

only collections with ≥ 20 rats (*i.e.*, 5b, 6c, 6g2, and 8f, see S1 Table). Finally, we ran PCR replicates on a subset of 94 *R. norvegicus* DNAs (30.2%) from the final dataset to check for allele scoring errors.

**Population genetic analyses: Determination of genetic groups and population assignment using STRUCTURE**

We determined the most likely number of genetic groups and conducted population assignment analyses using STRUCTURE v2.3.4 analysis [71] on the MSAT genotypes obtained from the 311 *R. norvegicus* DNAs. We set Bayesian parameters to assume genetic admixture and correlated allele frequencies in the study system, and to ignore collection sites. These parameters were run at $K$ values from 1–15, with 25,000 burn-in iterations followed by 100,000 run iterations. Bayesian analysis was repeated ten times for each $K$. Resulting log (Prob K) values were analyzed with the delta-$K$ method [72] to estimate the most likely number of genetic populations. We assigned rats to a specific genetic group and inferred potential migrants if their probability of membership ($Q$) to a specific genetic group was ≥ 0.75. Individual rats not meeting these criteria were grouped broadly into an "admixed" group, regardless of collection site (S3 Table). Using these results, we sorted all 311 rats according to their predicted genetic groups [12 distinct genetic groups (n = 247) and one admixed group (n = 64)] for additional population genetic analyses [~24% of individuals (64 admixed and 11 potential migrants, see Results) were no longer grouped according to their initial collection site].

**Population genetic analyses: Estimates of genetic structure at two scales.** We estimated overall genetic structure among Boston *R. norvegicus* using PCoA and three different estimators of $F_{ST}$. For PCoA, we used standardized genetic distances among all pairs of individuals in GenAlEx [68]. We then conducted *a posteriori* $F_{ST}$ analyses using only individuals from the 12 genetic groups identified in STRUCTURE (n = 247) but also removed juveniles (weight < 140g) [73] and inferred migrants, which could artificially increase or decrease, respectively, $F_{ST}$ estimates among the 12 predicted genetic groups. Collection 11b was the single exception because only juveniles were trapped and they appeared to form a distinct genetic group (Orange, see Results). These filters resulted in a final dataset for this analysis of 119 individual rats among the 12 genetic groups; all genetic groups were represented by ≥ 4 individuals (up to 19; S3 Table). We generated $F_{ST}$ estimates of theta ($\theta$) in FSTAT version 2.9.4 using Weir and Cockerham's method [74] to estimate population structure. Pairwise population $F_{ST}$ values were also estimated in FSTAT and *p*-values for all comparisons were generated using $a = 0.05$. We performed AMOVA on this same reduced set (n = 119) using standardized genetic distances in GenAlEx with 999 permutations to estimate PhiPT (an $F_{ST}$ analog). The third $F_{ST}$ estimator was a one-way global ANOSIM in PRIMER v5.2.9 (PRIMER-e, Albany, AK, NZ), conducted on a reduced set of 247 individual rats from the 12 distinct genetic groups (the 64 individuals forming the admixed group were removed but inferred migrants and juveniles were included; S3 Table); the R statistic was generated using 10,000 permutations. Afterwards, a post-hoc test was conducted to determine the genetic groups contributing to R, wherein pairwise comparisons among all genetic groups were conducted. The global alpha was divided by 66 (number of pairwise comparisons) to calculate a conservative Bonferroni corrected $a = 0.000758$.

**Population genetic analyses: Isolation by distance.** Because *R. norvegicus* relatives often live in close proximity, we performed a fine-scale analysis of spatial genetic structure (SGS) on all 311 rats (adults and juveniles) using spatial autocorrelation in GenAlEx with options set to SinglePOP, even distance classes (n = 20), and 999 permutations. Distance class intervals were increased in 100 m increments (0–100 m, 0–200 m, etc.) up to a maximum of 0–2,000 m. We assessed isolation by distance using the RELATE function in PRIMER v5.2.9 and standardized genetic distances and geographic locations (latitude and longitude) of the 247 rats (adults and juveniles) forming the 12 distinct genetic groups (S3 Table). We generated Spearman's rank correlation coefficient (Rho) using 10,000 permutations. Inbreeding coefficient $F_{IS}$ was estimated for each genetic group and global relatedness was estimated among all groups using the reduced set of 119 rats (no juveniles or inferred migrants) in FSTAT (S3 Table).

### *Leptospira* spp. genomics and genotyping

**Whole genome sequencing of isolates.** Whole genome sequencing of DNA extracted from 5 mL cultures of each isolate was performed as previously described [75,76]. Briefly, each isolate was propagated in HAN liquid media at 29°C, which was harvested by centrifugation at 10,000 x *g* for 15 min and DNA extracted using the Maxwell RSC Purefood Purification Pathogen kit. For Illumina library preparation, the concentration of reconstituted genomic DNA was determined with a Qubit dsDNA HS assay using a 3.0 fluorometer (Invitrogen, Carlsbad, CA, US). Whole-genome sequences were obtained (MiSeq Desktop Sequencer, 2x250 v2 paired-end chemistry and the Nextera XT DNA Library Preparation Kit, Illumina, San Diego, California, US) per manufacturer's instructions.

**Capture and enrichment of *Leptospira* spp. DNA from rat and human samples.** We attempted *Leptospira* spp. genome capture and enrichment from 22 PCR positive rat samples collected between 2017–2019, 20 culture negative and two culture positive (R316 and R47); the latter served as genome analysis controls. We also attempted enrichment on two separate human samples from Boston: one serum from 2017 (MA-1) and one concentrated urine from 2018 (MA-2). These sample DNAs were provided by the CDC and had *lipL32* PCR Ct values of 38.03 and 31.6, respectively. DNA from these 24 samples were subjected to pan pathogenic *Leptospira* spp. DNA capture and enrichment as previously described [28]. Briefly, sample DNAs were diluted separately to ~4 ng/µL in a volume of 55 µL, sonicated to an average size of 285 bp using a Q800R2 sonicator (QSonica, Newtown, CT, US), and then single indexed libraries were prepared using Agilent Sure-Select methodology (XT-HS kit, Agilent, Santa Clara, CA, US). The libraries (some pooled) were subjected to one or two rounds of DNA capture and enrichment (S5 Table) and sequenced on an Illumina MiSeq instrument using either a MiSeq v3 600 cycle (2x300 reads) or v2 300 cycle kit (2x150 reads). Three samples (R309, R334-19, and R379) failed to generate successful sequencing libraries and were excluded from DNA capture and enrichment and Illumina sequencing and six others failed to enrich *Leptospira* spp. DNA and were excluded from sequencing (R66, R67, R108, R318, R324 and MA-1) (S2 and S5 Tables). These unsuccessful attempts were associated with low abundance samples [PCR Ct values > 38 (n = 2)], enzymatic inhibition [failed library preparation (n = 3)], or perhaps contained divergent or non-pathogenic species of *Leptospira* spp. or some other genetically similar species not compatible with our pan pathogenic *Leptospira* spp. DNA capture and enrichment system (n = 4 including all three *Leptospira*-positive rats from collection 5b). Pooling schemes, rounds of DNA capture and enrichment, and Illumina sequencing kits used for each sample are provided in S5 Table.

### Read classifications, read mapping, SNP calling, and phylogenomics

To estimate the percentage of *Leptospira* spp. reads in enriched sequences and for taxonomic classification of enrichments and isolates, reads were mapped against the standard Kraken database with Kraken v2.1.2 [77]. To determine breadth of coverage for enriched genomes, reads were aligned against *L. interrogans* serovar Copenhageni strain Fiocruz L1-130 (GenBank accession# GCA_000007685.1) with minimap2 v2.22 [78] and the per base depth of coverage was calculated with Samtools v1.6 [79]. Two *Leptospira* spp. phylogenies were built. The first included a diverse set of *L. interrogans* (representing > 9 serovars) for the purpose of contextualizing Boston rat and human derived genomic diversity within *L. interrogans*. Single nucleotide polymorphisms were identified among 28 genomes from cultured isolates (GenBank accession numbers provided in phylogeny), 21 publicly available *L. interrogans* genomes, and 13 enriched genomes (12 rats and 1 human) by aligning reads against reference genome *L. interrogans* serovar Copenhageni strain Fiocruz L1-130 using minimap2 v2.22 and calling SNPs from the BAM file with GATK v4.2.2 [80] using a depth of coverage ≥ 3x and a read proportion of 0.9. SNPs that fell within duplicated regions, based on a reference self-alignment with NUCmer v3.1 [81], were filtered from downstream analyses. All methods were wrapped by NASP v1.2.1 [82]. A maximum likelihood phylogeny was inferred on the concatenated SNP alignments using IQ-TREE v2.2.0.3 with default parameters [83], the "-fast" option, and the integrated ModelFinder method [84]; the phylogeny was midpoint rooted. Second, to provide increased genomic resolution within *L. interrogans* and assess fine scale relatedness among the Boston genomes we replaced the

21 diverse *L. interrogans* genomes with two complete, publicly available *L. interrogans* serovar Copenhageni genomes (GenBank accession# GCA_002073495.2 and GCA_010978155.1). A maximum likelihood phylogeny was inferred on the concatenated SNP alignments using IQ-TREE v2.2.0.3 as described above, except with 1000 bootstrap replicates; the phylogeny was rooted with *L. interrogans* serovar Copenhageni strain Fiocruz L1-130.

**Leptospira interrogans reference phylogeny and WG-FAST placement of enriched samples.**  Two enriched samples (R289 and R292) had lower breadth and depth of coverage than the other enrichments (S5 Table) and were inserted into the phylogeny with WG-FAST v1.2 [85] using default settings. Reads were simulated from genome assemblies with ART vMountRainier [86] using the command "-p -na -ss MSv3 -l 250 -f 75 -m 300 -s 30". Based on the position of the enriched reads in the complete *L. interrogans* phylogeny (see Results), simulated reads were aligned against *L. interrogans* serovar Copenhageni strain Fiocruz L1-130 with minimap2 v2.24 [78] and SNPs were called with GATK v4.2.6.1 [80]. A maximum likelihood phylogeny was inferred on the concatenated SNPs with RAxML-NG v. 1.1.0 [87] and rooted with *L. interrogans* serovar Copenhageni strain Fiocruz L1-130.

**Human and rat phylogenomic comparison.**  To provide additional resolution among *Leptospira* spp. genomes derived from four *R. norvegicus* and one human that grouped together within a single *L. interrogans* serovar Copenhageni clade (designated as clade 2, see Results), we conducted a phylogenomic analysis using NASP v1.2.1 and IQ-TREE v2.2.0.3 as described above, except *L. interrogans* serovar Copenhageni strain SK1 (GenBank accession# GCA_010978155.1) was used as the reference and root.

## AmpSeq methods

We performed highly multiplexed amplicon sequencing (AmpSeq) on two subsets of rat samples that were PCR positive for *Leptospira* spp. but not included (or were unsuccessful) in the above steps for culture or genome enrichment. The first subset consisted of rats from 2016 and 2020–2022 that were PCR positive but culture negative (n = 11); we included one culture positive rat (Cad99) to serve as an analysis control. The final subset for AmpSeq contained any 2017–2019 positive samples that failed DNA capture and enrichment (n = 8 rats and n = 1 human) (S2 Table).

In total, 42 primer sets were designed and validated for the *Leptospira* spp. AmpSeq system. Thirteen primer sets were either modified from existing primer sets [60,88–98] targeting genes commonly used for molecular typing or detection of *Leptospira* spp. (*lipl32*, 16S, *secY*, *Loa22*, *OmpL1*, *flaB*, LIC_11108, LFB1, *ligB*), or redesigned from MLST loci (*glmU*, *lipL41*, *icdA*, *sucA*) [99–101] to be compatible with AmpSeq (153–415 bp in length) (S8 Table). We also designed 29 new primer sets targeting loci determined, via comparative genomic analyses, to be SNP-rich and highly discriminatory among 778 genome assemblies from 64 pathogenic and non-pathogenic *Leptospira* spp. (S9 Table). Assemblies were downloaded from the NCBI RefSeq database [102,103] using ncbi-genome-download v0.3.1 https://github.com/kblin/ncbi-genome-download/tree/0.3.1 and aligned to reference genome *L. interrogans* serovar Copenhageni strain Fiocruz L1-130 using NUCmer [104], and SNPs were identified using NASP v1.2.0 [82]. The SNPs were analyzed using VaST [105] to identify a set of targets maximizing discriminatory power among the assembled genomes. To avoid paralogous sequence variants and loci with highly variable alleles, we only analyzed SNPs located within a genus-wide core genome [106]. To maximize amplification of these targets across *Leptospira* spp., primers were designed with ambiguous bases for variants that occurred in ≥ 5% of genome assemblies.

To generate a conservative estimate of species coverage and discrimination for this AmpSeq system, we used *in silico* PCR to predict amplification for each of the 42 loci across 69 *Leptospira* spp. genome assemblies listed on the International Leptospirosis Society's resource webpage (accessed 15-Jan-2024; https://leptosociety.org/resources/); these genomes represent all validated pathogenic (P1 and P2) and saprophytic (S1 and S2) *Leptospira* spp. [57,107]. Genomes were downloaded with the ncbi-genome-download tool (https://github.com/kblin/ncbi-genome-download) from the RefSeq database [102] and renamed based on annotation information in the RefSeq record. PCR primers were screened against the 69 *Leptospira* spp. genomes with the -search_pcr2 method in USEARCH v11.0.667 [108] with maximum number of

SNP mismatches allowed in each primer set to two, using parameters: "-strand both -maxdiffs 2 -maxamp 500". Predicted genome hits were compiled for each primer set. This *in silico* analysis predicted amplification in all 69 *Leptospira* spp. The number of expected loci for each species varied, ranging from a single locus to all 42 loci. The pathogenic P1 clade was well represented with predicted amplification of 18–42 loci in all 20 species (S1 File).

AmpSeq was performed on two sample sets. The first set was used to validate the method and consisted of nine genomic DNA extracts obtained from isolates from seven P1 clade *Leptospira* spp. [*L. interrogans* (n = 2), *L. kirschneri*, *L. santarosai*, *L. noguchii* (n = 2), *L. borgpetersenii*, *L. alexanderi*, and *L. kmetyi*]. The second set contained unknown *Leptospira* spp. and consisted of DNA from 20 *Leptospira* PCR positive *R. norvegicus* kidney samples and one human serum sample (MA-1) (S2 Table) along with *L. interrogans* serovar Copenhageni strain Fiocruz L1-130 (ATCC, Manassas, Virginia US, catalog# BAA-1198D-5), which served as a PCR and analysis control, and also molecular grade water as a non-template negative control. The 42 loci were arranged into four multiplexes and PCRs were carried out in 10 μL volumes containing the following reagents (given in final concentrations): 10–20 ng of DNA template, 1x PCR buffer, 2.5 mM MgCl2, 0.2 mM dNTPs, 0.8 U Platinum Taq polymerase (Invitrogen, Carlsbad, CA, US), and 0.1-0.4 μM of each primer pair (see S8 Table). Thermocycling utilized the following conditions: 10 minutes at 95°C to release the Platinum Taq antibody, followed by 38 cycles of 60 s at 94°C, 30 s at 55°C and 30 s at 72°C. PCRs were pooled together in equal amounts in a final volume of 25 μL (6.25 μL of each multiplex) and sequence libraries were prepared for all positive samples as previously described [56,109]. Uniquely indexed sample libraries were pooled together in equimolar amounts and sequenced in parallel on an Illumina NextSeq instrument using a 600 cycle (2x300) P1 kit (Illumina, San Diego, CA, US).

## AmpSeq analysis

**Read mapping and classification.** Illumina sequencing reads for each individual sample were analyzed using the Amplicon Sequencing Analysis Pipeline (ASAP; https://github.com/TGenNorth/ASAP), wherein reads from each locus are mapped against a reference database using Bowtie2 and binned according to alignment score [110]; the number of paired end reads assigning to each reference are then counted. To build a comprehensive reference database of *Leptospira* spp. to be used for read mapping and classification, we implemented two strategies. First, we downloaded reference sequences from GenBank for the thirteen previously published loci and removed redundant sequences. Second, for the 29 new loci we bioinformatically extracted each locus from the reference set of 69 *Leptospira* spp. genome assemblies described above. We first amplified each locus separately (using the same PCR conditions above for the AmpSeq multiplexes) using gDNA from *L. interrogans* serovar Copenhageni strain Fiocruz L1-130. PCR products were visualized on a 2% agarose gel to ensure robust amplification and treated with ExoSAP-IT (Affymetrix, Santa Clara, CA, US) using 1 μL of ExoSAP-IT and 7 μL of PCR product under the following conditions: 37°C for 15 minutes followed by 80°C for 15 minutes. Treated products were then diluted (based on amplicon intensity) and sequenced using forward primer UT1F-acccaactgaatggagc and reverse primer UT2R-acgcacttgacttgtcttc (universal sequences incorporated into each unique primer pair, see S8 Table) in a Terminator v3.1 Ready Reaction Mix (Applied Biosystems, Foster City, CA, US). We used 10 μL volumes for sequencing reactions containing the following reagents (given in final concentrations): 5x Sequencing Buffer, 1 μL Terminator v3.1 Ready Reaction Mix, 1 μM primer, and 5 μL diluted PCR product. Thermocycling conditions were: 96°C for 20 seconds, followed by 30 cycles of 96°C for 10 seconds, 50°C for 5 seconds, and 60°C for 4 minutes. Sanger sequences were assembled, trimmed, and converted to fasta format using SeqMan Pro (DNASTAR Lasergene, Madison, WI, USA), which were used to extract each locus (if present) from the 69 *Leptospira* spp. genome assemblies using BLASTN [111] and a custom python script (https://gist.github.com/jasonsahl/2a232947a3578283f54c).

**Whole genome versus AmpSeq phylogenetic resolution.** We compared phylogenetic resolution and tree topology among two maximum likelihood phylogenies generated, respectively, from whole genome reads and AmpSeq reads. Using 24 genome assemblies representing 18 P1 clade pathogenic *Leptospira* spp. (GenBank accession numbers

provided in the phylogeny), we conducted a whole genome phylogenetic analysis using NASP v1.2.1 and IQ-TREE v2.2.0.3 as described above with 1000 bootstrap replicates and *L. interrogans* serovar Copenhageni strain L1-130 as the reference. We then conducted an identical analysis but also included the AmpSeq generated reads from the nine *Leptospira* spp. isolate genomic DNAs in our validation set. Inclusion of AmpSeq generated reads reduced the core genome to only nucleotide positions 1) included in the 42 loci, and 2) shared among the genome (n = 24) and AmpSeq (n = 9) reads. Both phylogenies were midpoint rooted.

**Species id and within species genotyping.** Rat and human DNAs that amplified at ≥ 7 loci (out of 42) with > 10x coverage were assigned a *Leptospira* spp. classification. To facilitate genotyping within species for samples with high breadth of coverage (*i.e.,* 25–42 loci amplified with > 10x coverage), SNPs were identified from AmpSeq reads and a reference set of 22 publicly available *L. interrogans* complete genome assemblies (GenBank accession numbers provided in the phylogeny). AmpSeq reads contain Illumina adaptors, universal tail sequences, and locus specific primers, all of which must be removed prior to SNP discovery if precise phylogenetic placement is the analysis goal; adaptors and universal tails were removed in ASAP and locus specific primers were trimmed using cutPrimers (https://github.com/aakechin/cutPrimers) [112]. Trimmed AmpSeq reads and genome assemblies were aligned against reference genome *Leptospira kirschneri* serovar Grippotyphosa strain RedPanda1 (GenBank accession# GCA_027563495.1) using NASP v1.2.1 [82]. A maximum likelihood phylogeny was then inferred on the concatenated SNP alignments using IQ-TREE v2.2.0.3 with the "-fast" option, default parameters [83], and the integrated ModelFinder method [84]; the phylogeny was rooted with *Leptospira kirschneri* serovar Grippotyphosa strain RedPanda1.

## Supporting information

**S1 Fig.** *Rattus norvegicus* **and pathogenic** *Leptospira* **detection summary plots.** A-F) *R. norvegicus* collections (larger circles) displayed by sampling year and color coded according to genetic group (S1 Table). Inferred migrants are represented by smaller circles and *lipL32* PCR positivity at 12 of 17 sites is indicated with asterisks. Major open spaces are indicated with green shading. G) Counts of total rats trapped (blue bars) and pathogenic *Leptospira* positive rats (orange bars) per year are displayed. H) The proportion of rats per year that were positive for pathogenic *Leptospira* spp. via *lipL32* PCR. The maps in panels A-F were created using ArcGIS software by Esri. ArcGIS and Arc-Map are the intellectual property of Esri and are used herein under license. Copyright Esri. All rights reserved. For more information about Esri software, please visit www.esri.com. Basemap: Light Gray Canvas Base https://www.arcgis.com/home/item.html?id=8b3d38c0819547faa83f7b7aca80bd76.
(TIF)

**S2 Fig. Plots from the delta-*K* method.** The hypothetical number of subpopulations (*K*) is plotted on the X-axis against A) delta-*K*, and B) The logarithm probability for each *K* [L(*K*)]. Ten independent runs in STRUCTURE were performed for each *K* value.
(TIF)

**S3 Fig. Principle Coordinates Analysis plot.** Principle Coordinates Analysis plot of standardized genetic distances illustrating genetic similarity among 311 *Rattus norvegicus* collected in Boston (2016–2022). Each rat is color coded according to its probability of membership (Q ≥ 0.75) to one of 12 genetic groups assigned using STRUCTURE (Fig 1 and S3 Table). The percentage of genetic variation explained by these two axes is indicated.
(TIF)

**S4 Fig. Spatial autocorrelation analysis graph.** Spatial autocorrelation analysis graph assessing the correlation between genetic and geographic distance (Y-axis) using twenty even distance classes corresponding to 100 m intervals (X-axis). The blue line is the correlation value (r), and the red dashed lines are the 95% confidence intervals (U = upper

and L=lower) about the null hypothesis of no spatial structure as determined by bootstrap resampling. Values of r above the upper 95% confidence interval are significant.
(TIF)

**S5 Fig.** *Leptospira interrogans* **whole genome phylogeny.** A maximum likelihood midpoint rooted phylogeny of 28 *Leptospira interrogans* serogroup Icterohaemorrhagiae isolate genomes and 13 enriched genomes from 40 rats and one human (bold text), together with 22 publicly available *L. interrogans* isolate genomes representing > 9 serovars (GenBank accession numbers are provided in the figure annotations), based upon a concatenated SNP alignment of 27,678 positions out of a core genome size of 1,186,500 nucleotides (nts).
(TIF)

**S6 Fig.** *Leptospira interrogans* **serogroup Icterohaemorrhagiae WG-FAST phylogeny.** A maximum likelihood phylogeny of 28 *Leptospira. interrogans* serogroup Icterohaemorrhagiae isolate genomes from rats, together with two publicly available *L. interrogans* serogroup Icterohaemorrhagiae serovar Copenhageni complete genomes (GenBank accession# GCA_002073495.2 and GCA_010978155.1), and 13 enriched genomes (12 rats and 1 human), based upon a concatenated SNP alignment of 127 positions out of a core genome size of 2,123,028 nucleotides (nts). The phylogeny was rooted with reference genome *L. interrogans* serogroup Icterohaemorrhagiae serovar Copenhageni strain Fiocruz L1-130 (GenBank accession# GCA_000007685.1). The color of the genome ID corresponds to the rat genetic group in Fig 1; black text indicates that no genetic group was assigned. Two enriched samples (R289 and R292) had lower breadth and depth of coverage than the other enrichments (see S5 Table) and were inserted into the rat phylogeny based upon canonical SNPs with WG-FAST.
(TIF)

**S7 Fig.** **Pathogenic** *Leptospira* **spp. whole genome and AmpSeq phylogenies.** A) Whole genome maximum likelihood midpoint rooted phylogeny including 18 P1 clade pathogenic *Leptospira* spp. B) compared to an AmpSeq phylogeny that also includes AmpSeq data generated from nine *Leptospira* spp. isolate gDNAs (blue text). Species specific clades were maintained (color coded) and similar phylogenetic topologies were observed despite a large reduction in the core-genome of the AmpSeq tree (1,195 nts) compared to the whole genome tree (167,849 nts). Bootstrap values are indicated at specific nodes. The nine AmpSeq *Leptospira* spp. isolate gDNAs grouped in their respective species clades.
(TIF)

**S8 Fig.** *Leptospira interrogans* **AmpSeq phylogeny.** Maximum likelihood phylogeny based upon an alignment of 4,260 shared nucleotide positions using *Leptospira* spp. AmpSeq data generated from nine *lipL32* qPCR positive *Rattus norvegicus* (bold text) along with 22 *Leptospira interrogans* isolate genomes (GenBank accession numbers are provided in the figure annotations). The phylogeny is rooted with *Leptospira kirschneri* serovar Grippotyphosa strain RedPanda1 (GenBank accession# GCA_027563495.1).
(TIF)

**S1 Table.** *Rattus norvegicus* **collections.**
(XLSX)

**S2 Table.** **Analyses summary and results for all rodent and human samples.**
(XLSX)

**S3 Table.** *Rattus norvegicus* **population assignment.**
(XLSX)

**S4 Table.** *Rattus norvegicus* **FSTAT results.**
(XLSX)

**S5 Table. DNA capture and enrichment setup and results.**
(XLSX)

**S6 Table. *Leptospira* spp. AmpSeq results.**
(XLSX)

**S7 Table. *Rattus norvegicus* trapping locations.**
(XLSX)

**S8 Table. *Leptospira* spp. AmpSeq primers and conditions.**
(XLSX)

**S9 Table. *Leptospira* spp. AmpSeq primer design genomes.**
(XLSX)

**S1 File. Predicted amplification of 42 AmpSeq loci against 69 *Leptospira* spp.**
(XLSX)

## Acknowledgments

The authors extend their gratitude to the City of Boston's Inspectional Services Department and the Boston Public Health Commission for their assistance. The authors thank the *Mycobacteria* & *Brucella* Section from the NVSL for their technical assistance in genome sequencing.

USDA is an equal opportunity provider and employer. Mention of trade names or commercial products in this publication is solely for the purpose of providing specific information and does not imply recommendation or endorsement by the U.S. Department of Agriculture. The content is solely the responsibility of the authors and does not necessarily represent the official views of the NIH. Any opinions, findings, and conclusions or recommendations expressed in this material are those of the authors and do not necessarily reflect the views of NSF. The findings and conclusions in this report are those of the authors and do not necessarily represent the official position of the Centers for Disease Control and Prevention.

## Author contributions

**Conceptualization:** Jarlath E. Nally, David M. Wagner, Marieke H. Rosenbaum.

**Funding acquisition:** Karen LeCount, Talima Pearson, Jarlath E. Nally, David M. Wagner, Marieke H. Rosenbaum.

**Investigation:** Nathan E. Stone, Camila Hamond, Joel R. Clegg, Ryelan F. McDonough, Reanna M. Bourgeois, Rebecca Ballard, Natalie B. Thornton, Marianece Nuttall, Hannah Hertzel, Tammy Anderson, Ryann N. Whealy, Skylar Timm, Alexander K. Roberts, Verónica Barragán, Wanda Phipatanakul, Jessica H. Leibler, Hayley Benson, Aubrey Specht, Ruairi White, Karen LeCount, Tara N. Furstenau, Renee L. Galloway, Nichola J. Hill, Joseph D. Madison, Viacheslav Y. Fofanov, Jason W. Sahl, Zachary Weiner, Jarlath E. Nally, David M. Wagner, Marieke H. Rosenbaum.

**Methodology:** Nathan E. Stone, Camila Hamond, Ruairi White, Tara N. Furstenau, Renee L. Galloway, Nichola J. Hill, Joseph D. Madison, Viacheslav Y. Fofanov, Jason W. Sahl, Jarlath E. Nally, David M. Wagner, Marieke H. Rosenbaum.

**Project administration:** Nathan E. Stone, Talima Pearson, Joseph D. Busch, Zachary Weiner, Jarlath E. Nally, David M. Wagner, Marieke H. Rosenbaum.

**Resources:** Ruairi White, Karen LeCount.

**Supervision:** Nathan E. Stone, Karen LeCount, Jason W. Sahl, Joseph D. Busch, Jarlath E. Nally, David M. Wagner, Marieke H. Rosenbaum.

**Validation:** Nathan E. Stone, Camila Hamond, Joel R. Clegg, Ryelan F. McDonough, Reanna M. Bourgeois, Rebecca Ballard, Tammy Anderson, Ryann N. Whealy.

**Visualization:** Nathan E. Stone.

**Writing – original draft:** Nathan E. Stone, Camila Hamond, Tara N. Furstenau, Joseph D. Busch, Jarlath E. Nally, David M. Wagner, Marieke H. Rosenbaum.

**Writing – review & editing:** Nathan E. Stone, Camila Hamond, Tammy Anderson, Karen LeCount, Joseph D. Madison, Jason W. Sahl, Joseph D. Busch, Jarlath E. Nally, David M. Wagner, Marieke H. Rosenbaum.

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
