## [Decision Letter · Decision Letter 0]

13 Sep 2024

Dear Dr. Rosenbaum,

Thank you very much for submitting your manuscript "Host population structure and rare dispersal events drive leptospirosis transmission patterns among Rattus norvegicus in Boston, Massachusetts, US" for consideration at PLOS Neglected Tropical Diseases. As with all papers reviewed by the journal, your manuscript was reviewed by members of the editorial board and by several independent reviewers. In light of the reviews (below this email), we would like to invite the resubmission of a significantly-revised version that takes into account the reviewers' comments.

We cannot make any decision about publication until we have seen the revised manuscript and your response to the reviewers' comments. Your revised manuscript is also likely to be sent to reviewers for further evaluation.

Sincerely,

Claudia Munoz-Zanzi

Guest Editor

Mathieu Picardeau

Section Editor

Reviewer's Responses to Questions

**Key Review Criteria Required for Acceptance?**

**Methods**

-Are the objectives of the study clearly articulated with a clear testable hypothesis stated?

-Is the study design appropriate to address the stated objectives?

-Is the population clearly described and appropriate for the hypothesis being tested?

-Is the sample size sufficient to ensure adequate power to address the hypothesis being tested?

-Were correct statistical analysis used to support conclusions?

-Are there concerns about ethical or regulatory requirements being met?

Reviewer #1: (No Response)

Reviewer #2: -Are the objectives of the study clearly articulated with a clear testable hypothesis stated?

Yes

-Is the study design appropriate to address the stated objectives?

Unclear. There is a tremendous amount of work presented in this manuscript, though I found the ultimate test of the stated hypothesis (does population structure of lepto match the structure of rat populations) more qualitative than quantitative. I would have expected a more quantitative analysis of spatial relationships of lepto isolates akin to what was performed for rats - if that had been performed, then I think a qualitative comparison of patterns would be acceptable inference. Instead, it appears that spatial structure of lepto isolates was based on the geographic placement of the rodents (or their identified cluster) from which they were isolated. Perhaps I missed a more explicit test of spatial relationships (?). At a minimum, spatial analyses of lepto isolates needs much more clarification than given.

-Is the population clearly described and appropriate for the hypothesis being tested?

Yes.

-Is the sample size sufficient to ensure adequate power to address the hypothesis being tested?

I believe so, though sampling effort is quite unequal and after following references, collected for different purposes based on the sampling time frame. Samples from 2021 - 2022 most closely match the stated hypothesis of the work. This should be discussed more thoroughly in the report.

-Were correct statistical analysis used to support conclusions?

Yes

-Are there concerns about ethical or regulatory requirements being met?

No, though I fail to see the relevance of the pathogenic work performed in hamsters. I would perhaps recommend removing this section in its entirety.

**Results**

-Does the analysis presented match the analysis plan?

-Are the results clearly and completely presented?

-Are the figures (Tables, Images) of sufficient quality for clarity?

Reviewer #1: (No Response)

Reviewer #2: -Does the analysis presented match the analysis plan?

Unclear. See comments above regarding methods.

-Are the results clearly and completely presented?

Mostly. There are so many things reported that it is incredibly difficult to synthesis all of the work into a single, comprehensive report. See editorial comments below.

-Are the figures (Tables, Images) of sufficient quality for clarity?

I think too much information is packed into the figures. I would have liked to see more summary plots of sampling results (such as location/number/lepto positive samples per year). Anything to more clearly show a temporal/spatial trend that isn't tied to the complex genetic results that are presented. Could even be included as supporting material.

**Conclusions**

-Are the conclusions supported by the data presented?

-Are the limitations of analysis clearly described?

-Do the authors discuss how these data can be helpful to advance our understanding of the topic under study?

-Is public health relevance addressed?

Reviewer #1: (No Response)

Reviewer #2: -Are the conclusions supported by the data presented?

No. I found the methods to identify dispersal inaccurate and I think the tie between spatial composition of lepto and rat populations could be more robust. At most, the data suggest relationships, but I am not convinced the relationships are as robust as stated.

-Are the limitations of analysis clearly described?

No. Many other animals can transmit lepto, correct? Movement of pathogens by other reservoirs is not discussed.

-Do the authors discuss how these data can be helpful to advance our understanding of the topic under study?

Yes.

-Is public health relevance addressed?

Yes.

**Editorial and Data Presentation Modifications?**

Reviewer #1: (No Response)

Reviewer #2: See comments above regarding spatial analyses of lepto isolates.

In terms of the term "dispersal", what is stated as "observed" is merely "inferred". As best as I can understand, dispersal is identified by individuals from a certain location clustering with individuals from a different location at a set Q threshold.

Related to dispersal, I think more caution needs to be used in especially in the context of lepto isolates. I infer from the paper that rats are driving all movement of lepto, which plausible, is complicated by the fact the other animals can be reservoirs of lepto. Movement by other reservoirs could wash out genetic signatures of the pathogens or provide a level of background noise that at least needs to be addressed in the discussion.

I think more consideration of K=2 should be discussed. Given sites 1 - 16 have much shorter spatial distances between them, just from looking at the map site 17 could be driving a lot of those spatial patterns. Did the authors look at the K=2 groupings

**Summary and General Comments**

Reviewer #1: Study overview

This is an important, well written and a “very long” paper that describes the potential influence of rat population dispersal in urban environment on the transportation and diversity of Leptospira. The authors reported approximately 18% of Leptospira-positive samples from the tested samples, representing 12 of the 17 study sites. Despite evidence of high genetic structure, the study found a limited dispersal rate in the urban study sites. The authors’ conclusion that the transportation and persistence of Leptospira in the environment are driven primarily by rat dispersal is questionable, given previous studies have indicated that adverse weather events such as flooding or even intense rainfall also significantly contribute to the same conclusion.

Additionally, the manuscript would benefit from addressing the following specific comments.

Comments to authors

• Line 81: I would change "such as" to "mainly"

• Line 84: Consider changing “leptospirosis” to "the presence of Leptospira …”

• Line 85: Change "movement" here to "transportation"

• Line 101: I will say "... other similar urban settings" here

• Lines 288 & 289: Could the author clarify how trapping efforts were accounted for here and also in the method, this should better reflect the population structure and dynamics than just relying on the number of rats captured at each site

• Lines 412-414: It would be nice to state what the authors think might be responsible for this long-distance migration, especially even among adult females which is quite strange. Additionally, please consider including how many of the migrant rats tested positive for Leptospira. This information could be valuable for understanding the role of rat dispersal in the spread/transportation of Leptospira pathogen

• Lines 415&416: Did the authors record any short-distance migration at these two sites, and for how long was sampling conducted at each site? This information is crucial given the distance among sites in this study and the fact that rats typically do not travel beyond 100m in urban environment. Without this information, it may somewhat represent a limitation of the study

• Lines 422 & 423: Consider to include how the rats were classified into juvenile and adult

• Lines 447 – 449: Could this also be associated with resource availability?

• Lines 473-475: Please state how many of the positive rats were migrants

• Lines 562-566: The authors have not sufficiently supported their conclusion here. As mentioned earlier, it is essential to characterize how many of the migrant rats tested positive for Leptospira. Without this information, it is difficult to rule out alternative explanations, such as the transportation of L interrogans serogroup Icterohaemorrhagiae genotypes by flooding after rainfall which is common in literature

• Line 645: group that ...?

• Lines 646 & 647: This statement requires cautions, in order not to over-blow the results

• Lines 662 & 663: For how many nights were the traps placed at each site, and what were the criteria for placing specific number of traps at each site?

• Line 673-675: Please include the procedure used for animal identification here

• Line 682 & 683: Add “pathogenic” before Leptospira

• Line 693: The authors should consider including a table with the sequences of the primers and probes used

• Line 694: Briefly describe the master mix reaction protocol with the concentrations of each reagent, the amount of DNA and the cycling program

• Lines 736-738: This way, on the average, how many samplings were conducted per year? Would it not be more appropriate to use term “sampling campaign” instead of the current term for clarity? The use of “collections” including in other parts of the manuscript can be confusing, as it may be interpreted as referring to the number of animals captured/collected. Please consider revising this terminology throughout the manuscript

• Line 757: Also, change "movement" to "transportation" here

• Lines 757-759: Considering the authors aims here, consider stating the frequency of trapping during the period of intensive sampling stated in Ln746 across the 3 sites. Also, briefly state the baiting and trapping protocol and for how long these lasted

• Line 765: Where or were?

• Lines 858-860: Considering that juvenile rats often wander more actively than matured rats ("Macdonald, D. W., Mathews, F., Berdoy, M., Singleton, G. H., & Leirs, H. (1999). Ecologically Based Rodent Management. The behaviour and ecology of Rattus norvegicus: from opportunism to kamikaze tendencies. Canberra: Australian Centre for International Agricultural Research, 49-80" and "https://www.researchgate.net/publication/353608762_Using_Rhodamine_B_to_Assess_the_Movement_of_Small_Mammals_in_an_Urban_Slum"), will the exclusion of juvenile rats not impact the result here, as their inclusion could provide insights into the overall population genetic diversity as oppose to just the genetic structure of only the breeding population?

Reviewer #2: This paper is essentially presenting three separate projects: 1) a longitudinal study of rats/lepto in boston, 2) a hyper-focused study of rat populations on either side of a suspected dispersal barrier, and 3) the development of new(er) technologies to isolate, amplify, and genotype lepto isolates. All three have merit on their own (though perhaps in different journals). Together, the main messages are lost, objectives get tied together when maybe they shouldn't be, and ultimately the stated hypothesis only matches a small subset of the sampling methodology (the 2021 - 2022 sampling). It reads as is that hyper-focused study was the ultimate driver of this work which is then supplemented by all of this other prior work that took place. I don't think there is anything wrong with that, but a lot of work more work needs to be put in to this manuscript to streamline what the ultimate objectives are. It could be the case that a lot of the methodology gets dumped into supporting information as it is ancillary to the objective of comparing population structure of lepto and rats.

PLOS authors have the option to publish the peer review history of their article (what does this mean? ). If published, this will include your full peer review and any attached files.

**Do you want your identity to be public for this peer review?** For information about this choice, including consent withdrawal, please see our Privacy Policy .

Reviewer #1: No

Reviewer #2: No
---

## [Decision Letter · Decision Letter 1]

23 Jan 2025

Please submit your revised manuscript within 30 days Feb 22 2025 11:59PM. If you will need more time than this to complete your revisions, please reply to this message or contact the journal office at plosntds@plos.org. Please include the following items when submitting your revised manuscript:

Response to Reviewers
Revised Manuscript with Track Changes
Manuscript

Section Editor

Shaden Kamhawi

co-Editor-in-Chief

Paul Brindley

co-Editor-in-Chief

**Journal Requirements:**

We do not publish any copyright or trademark symbols that usually accompany proprietary names, eg ©,  ®, or TM  (e.g. next to drug or reagent names). Therefore please remove all instances of trademark/copyright symbols throughout the text, including:

- ® on page: 44.

**Reviewers' comments:**

**Key Review Criteria Required for Acceptance?**

**Methods**

-Are the objectives of the study clearly articulated with a clear testable hypothesis stated?

-Is the study design appropriate to address the stated objectives?

-Is the population clearly described and appropriate for the hypothesis being tested?

-Is the sample size sufficient to ensure adequate power to address the hypothesis being tested?

-Were correct statistical analysis used to support conclusions?

-Are there concerns about ethical or regulatory requirements being met?

Reviewer #1: Partly yes

Reviewer #2: (No Response)

**Results**

-Does the analysis presented match the analysis plan?

-Are the results clearly and completely presented?

-Are the figures (Tables, Images) of sufficient quality for clarity?

Reviewer #1: Partly yes

Reviewer #2: (No Response)

**Conclusions**

-Are the conclusions supported by the data presented?

-Are the limitations of analysis clearly described?

-Do the authors discuss how these data can be helpful to advance our understanding of the topic under study?

-Is public health relevance addressed?

Reviewer #1: Yes

Reviewer #2: (No Response)

**Editorial and Data Presentation Modifications?**

Reviewer #1: (No Response)

Reviewer #2: (No Response)

**Summary and General Comments**

Reviewer #1: I appreciate the authors for revising their manuscript and incorporating the reviewers’ comments. However, I still find the manuscript somewhat lengthy and a bit difficult to follow, as it attempts to encompass a large amount of information. It would be beneficial if the authors could rationalize the content to focus more closely on their primary study objective.

Otherwise, the manuscript shows significant improvement from the previous version. I have included a few minor comments that may help further enhance the manuscript.

• Kindly review and correct the manuscript throughout. Rats do not transport leptospirosis but rather the pathogen (Leptospira spp.), while the disease being transmitted is leptospirosis. I noticed that these two terms are occasionally used incorrectly in the manuscript.

• The phrase “6-10 trapping units were placed at each site...” is unclear. Did you mean that each site consisted of 6-10 trapping units, or that 6-10 traps were placed at each site? Additionally, the description “the trapping unit remained in place…” requires critical revision. Please provide a clear and concise explanation of what you mean by “trapping unit”, “trapping sites” and “trapping location” at the first mention to aid readers’ understanding.

• Please cite a reference for this statement “Carcasses were visually identified as rats by the study team based on morphology and morphometry, and Rattus norvegicus is the only species of rat currently inhabiting Massachusetts”.

• Consider moving lines 736-742 under the study location and rodent trapping. You could further separate this sub-heading into “study area/description of study location” and “rodent trapping”. Then you can move the reasons for intensively sampling the selected sites up, so that the readers understand the rationale without having to wait to read several lines. It would be helpful to explain why the three areas were intensively sampled during the stated periods and not at another time.

Reviewer #2: (No Response)

PLOS authors have the option to publish the peer review history of their article (what does this mean? ). If published, this will include your full peer review and any attached files.

**Do you want your identity to be public for this peer review?** For information about this choice, including consent withdrawal, please see our Privacy Policy .

Reviewer #1: No

Reviewer #2: No

**Figure resubmission:**

**Reproducibility:** To enhance the reproducibility of your results, we recommend that authors of applicable studies deposit laboratory protocols in protocols.io, where a protocol can be assigned its own identifier (DOI) such that it can be cited independently in the future. Additionally, PLOS ONE offers an option to publish peer-reviewed clinical study protocols. Read more information on sharing protocols at https://plos.org/protocols?utm_medium=editorial-email&utm_source=authorletters&utm_campaign=protocols

---

## [Editor Report · Decision Letter 2]

5 Mar 2025

Dear Dr. Rosenbaum,

We are pleased to inform you that your manuscript 'Host population dynamics influence Leptospira spp. transmission patterns among Rattus norvegicus in Boston, Massachusetts, US' has been provisionally accepted for publication in PLOS Neglected Tropical Diseases.

Best regards,

Stuart D. Blacksell

Section Editor

Shaden Kamhawi

co-Editor-in-Chief

Paul Brindley

co-Editor-in-Chief

---

## [Editor Report · Acceptance letter]

Dear Dr. Rosenbaum,

We are delighted to inform you that your manuscript, "Host population dynamics influence *Leptospira* spp. transmission patterns among Rattus norvegicus in Boston, Massachusetts, US," has been formally accepted for publication in PLOS Neglected Tropical Diseases.

Best regards,

Shaden Kamhawi

co-Editor-in-Chief

Paul Brindley

co-Editor-in-Chief
